# Enzymatic β-elimination in natural product *O*- and *C*-glycoside deglycosylation

Johannes Bitter[1], Martin Pfeiffer[1], Annika J. E. Borg [1,2], Kirill Kuhlmann[3], Tea Pavkov-Keller [3,4,5], Pedro A. Sánchez-Murcia [6] & Bernd Nidetzky [1,2] ✉

Biological degradation of natural product glycosides involves, alongside hydrolysis, β-elimination for glycosidic bond cleavage. Here, we discover an *O*-glycoside β-eliminase (OGE) from *Agrobacterium tumefaciens* that converts the C3-oxidized *O*-β-D-glucoside of phloretin (a plant-derived flavonoid) into the aglycone and the 2-hydroxy-3-keto-glycal elimination product. While unrelated in sequence, OGE is structurally homologous to, and shows effectively the same Mn²⁺ active site as, the *C*-glycoside deglycosylating enzyme (CGE) from a human intestinal bacterium implicated in β-elimination of 3-keto *C*-β-D-glucosides. We show that CGE catalyzes β-elimination of 3-keto *O*- and *C*-β-D-glucosides while OGE is specific for the *O*-glycoside substrate. Substrate comparisons and mutagenesis for CGE uncover positioning of aglycone for protonic assistance by the enzyme as critically important for *C*-glycoside cleavage. Collectively, our study suggests convergent evolution of active site for β-elimination of 3-keto *O*-β-D-glucosides. *C*-Glycoside cleavage is a specialized feature of this active site which is elicited by substrate through finely tuned enzyme-aglycone interactions.

Flavonoids are a large group of polyphenol natural products[1-3] that are often biosynthesized as glycosides[4-7]. Contrasting the structural diversity of flavonoids, their glycosylation is generally simple, consisting of a single β-D-glucosyl residue attached to the polyphenol core[8,9]. Phenolic oxygens are the preferred glycosylation sites. In rare instances, the glycosylation also happens at an aromatic carbon (Fig. 1a)[8-11]. Flavonoid *C*-glycosides differ in chemical properties and bioactivities from the corresponding *O*-glycosides[10,11]. Flavonoid uptake in humans often necessitates that the glycoside is cleaved to release the free aglycone[12-16]. The enzymes of glycoside cleavage derive from the body's own repertoire (e.g., epithelial glycoside hydrolase[17,18]) or are from the human gut microbiome[19-23].

The *C*-glycosides pose fundamental challenge for cleavage because the canonical enzyme mechanisms of glycoside hydrolysis (Fig. S1)[24] fail with these substrates[25-27]. But even with flavonoid *O*-glycosides[18,28], there arises conundrum for the glycoside hydrolase to work as a generalist catalyst of bond cleavage in β-D-glucoside substrates that represent a broad diversity of aglycone structures. The glycosides are typically resistant to spontaneous hydrolysis. Glycoside hydrolases are among the most proficient enzymes known, with rate accelerations of up to $10^{17}$ provided to the chemical step[24,29]. In context of natural product glycosides, the interesting question arises whether a different mechanism of biological catalysis, that is, one that subdivides the difficult task of glycosidic bond cleavage into more easily manageable steps, could represent a viable alternative[30,31].

The current research is built on the seminal discovery of enzymes that catalyze glycosidic bond cleavage via β-elimination of the suitably activated (i.e., 3-keto-pyranosyl) substrate. The Withers group identified enzymes of glycoside hydrolase family GH4 to use a four-step mechanism, comprised of oxidation by enzyme-NAD⁺, β-elimination,

¹Institute of Biotechnology and Biochemical Engineering, Graz University of Technology, NAWI Graz, Petersgasse 12, A-8010 Graz, Austria. ²Austrian Centre of Industrial Biotechnology, Krenngasse 37, A-8010 Graz, Austria. ³Institute of Molecular Biosciences, University of Graz, NAWI Graz, Humboldtstraße 50/III, A-8010 Graz, Austria. ⁴BioTechMed-Graz, Mozartgasse 12/II, A-8010 Graz, Austria. ⁵BioHealth Field of Excellence, University of Graz, Humboldtstraße 50, A-8010 Graz, Austria. ⁶Laboratory of Computer-Aided Molecular Design, Division of Medicinal Chemistry, Otto-Loewi Research Center, Medical University of Graz, Neue Stiftingstalstraße 6/III, A-8010 Graz, Austria. ✉e-mail: bernd.nidetzky@tugraz.at

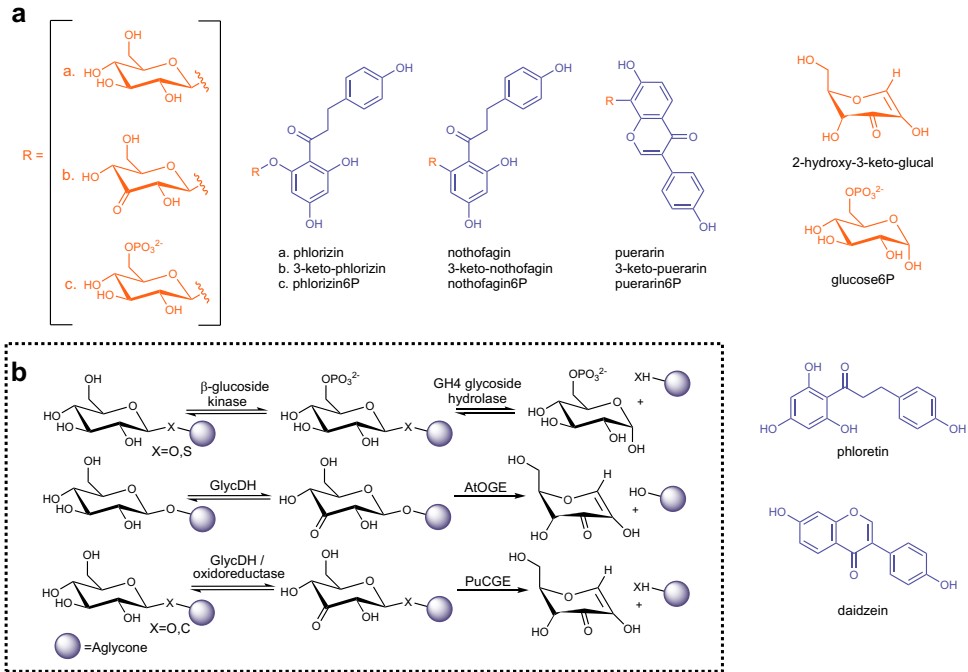

**Fig. 1 | Glycoside cleavage by oxidation and elimination. a** Glycoside substrates and their cleavage products. **b** Enzymatic paths of glycoside cleavage by β-elimination. C3 oxidation catalyzed by dehydrogenase GlycDH[46,47] or an oxidoreductase[44], phosphorylation by a β-glucoside kinase[89]. Deglycosylation catalyzed by GH4 glycoside hydrolases[30], or eliminases *Pu*CGE[41,45] (this work) and *At*OGE (this work). Nomenclature: 3-keto-phlorizin; 3,5-dihydroxy-2-[3-(4-hydroxyphenyl)-1-oxopropyl]phenyl β-D-*ribo*-hexopyranosid-3-ulose; 3-keto-puerarin, 8-(β-D-*ribo*-hexopyranos-3-ulos-1-yl)-7-hydroxy-3-(4-hydroxyphenyl)-4H-1-benzo-pyran-4-one; 3-keto-nothofagin, 1-(3-(β-D-*ribo*-hexopyranos-3-ulos-1-yl)−2,4,6-tri-hydroxyphenyl)−3-(4-hydroxyphenyl)−1-propanone; 6P = 6-phosphate.

water addition to the anomeric center, and reduction by enzyme-NADH, to cleave the glycosidic bond in 6-phospho-hexopyranosyl substrates (Fig. 1b; details in Fig. S2)[30–35]. The catalytic reaction proceeds with *O*- but also *S*-glycosides[36] and the mechanism has been explored deeply[32,33,35,37,38]. Family GH109[39,40] glycoside hydrolases also use elimination chemistry linked to transient oxidation for glycosidic bond cleavage.

Based on studies of the flavonoid *C*-β-glucoside puerarin (Fig. 1), the Iwashima and Hattori group[41–43] showed a multienzyme pathway of substrate conversion in the human intestinal bacterium PUE, a related species of *Dorea longicatena* LCR19. As a key step, this pathway involves β-elimination of the 3-keto-substrate for glycosidic bond cleavage and release of the daidzein aglycone (Fig. 1b; details in Fig. S3)[44]. The immediate elimination product (2-hydroxy-3-keto-D-glucal; 1,5-anhydro-D-*erythro*-hex-1-en-3-ulose) is converted to 3-keto-D-glucose which is then reduced to D-glucose (Fig. S3)[44]. As shown by the Kobayashi group, the crucial β-eliminase is a heterodimeric Mn[2+] metalloenzyme[45]. The enzyme is naturally present in various species of the human gut microbiome and is also broadly distributed within soil and marine bacteria[45]. Later in the Discussion, we return to the interesting question whether such widespread occurrence of the *C*-glycoside deglycosylating enzymes is reasonably in accord with the natural scarcity of the actual substrates. In their mechanistic proposal for the β-eliminase reaction (Fig. S4), Mori et al.[45] emphasize an energetically demanding dearomatization of the flavonoid aglycone as central to the *C-C* bond cleavage, different fundamentally from the *O-C* bond cleavage. The evidence of the current study suggests β-eliminases of *C*- and *O*-glycoside deglycosylation united by common features of active-site structure and catalytic mechanism used. Cleavage of the *C*-glycosidic bond is thus understood as a specialty of the conserved enzyme mechanism, requiring the addition of specific positioning, potentially including general-base catalytic activation, of the aglycone to the core active-site machinery of the β-eliminase. To distinguish between β-

eliminases cleaving *O*- and *C*-glycosides, we henceforth refer to them with the abbreviations OGE and CGE.

Here, we show that the flavoenzyme dehydrogenase GlycDH, known from earlier work to catalyze regioselective oxidation of natural product *O*-glucosides[46,47], also oxidizes *C*-glucosides with broad specificity. From the GlycDH genomic context in *Agrobacterium tumefaciens*, we thus identify a multienzymatic system of glycoside cleavage. Strikingly, the *A. tumefaciens* systems conserves the biochemical pathway of the human intestinal bacterium PUE[41], but the enzymes used are completely different. In particular, there is the β-eliminase *At*OGE discovered in here. Despite sequence similarity lacking entirely, *At*OGE and CGE are structurally homologous and show effectively the same Mn[2+] containing active site. *Pu*CGE (CGE from the gut intestinal bacterium PUE) is shown to catalyze the β-elimination of both 3-keto *O*- and *C*-β-D-glucosides whereas *At*OGE is specific for the *O*-glycoside. Substrate comparisons and mutagenesis reveal the requirements for 3-keto *C*-β-D-glucoside reactivity of *Pu*CGE lacking in *At*OGE. Family GH4 glycoside hydrolases are inactive with the relevant 6-phospho-derivatives of puerarin and nothofagin (Fig. 1). They cleave the 6-phospho *O*-β-D-glucoside of phloretin (Fig. 1), yet at greatly decreased efficiency (≥10[5]-fold) compared to *At*OGE and *Pu*CGE cleaving the corresponding (non-phosphorylated) 3-keto-phlorizin substrate (Fig. 1). Collectively, our study shows the convergent evolution of a β-eliminase active site for the cleavage of 3-keto *O*-β-D-glucosides in biological pathways of natural product deglycosylation. The *C*-glycoside cleavage is revealed as a specialized feature of this general active site. Enzymatic reactivity towards *C*-glycosides requires actuation by the substrate through finely tuned enzyme-aglycone interactions.

## Results

### GlycDH catalyzes C3 oxidation of flavonoid *C*-glucosides

GlycDH is a bacterial flavoenzyme oxidoreductase, originally identified from *Rhizobium* sp. GIN611[46]. The enzyme catalyzes C3

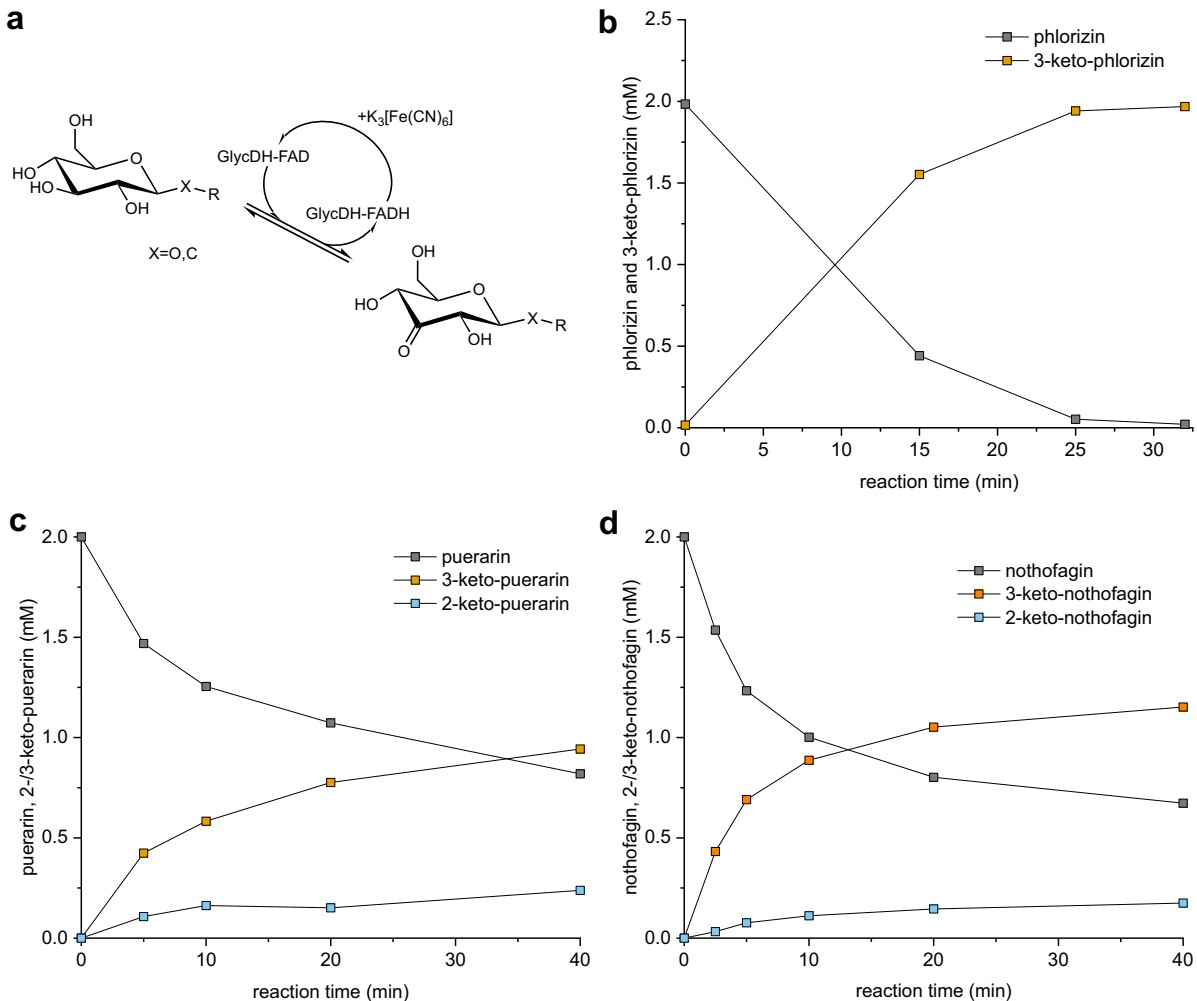

**Fig. 2 | GlycDH catalyzed oxidation of *O*- and *C*-β-ᴅ-glucosides. a** Reaction scheme. **b–d** Time courses of reactions with phlorizin (**b**), puerarin (**c**), and nothofagin (**d**). Conditions: 100 mM potassium phosphate, pH 5.7 (**b**, **d**), with 2.0 mM TCEP added (**c**). The substrate (2.0 mM) was incubated with K$_3$[Fe(CN)$_6$] (8.0 mM), DMSO (5.0 vol.%) and GlycDH (0.02, 0.25 and 0.10 mg/mL for **b–d**, respectively (*n* = 1 individual experiment). Source data are provided as a Source data file.

oxidation of *O*-glycosides with broad specificity[47]. Here we demonstrate flavonoid *C*-β-ᴅ-glucosides (puerarin, nothofagin, Fig. 1; orientin, vitexin, isovitexin, Fig. S5) as substrates of the GlycDH. The *O*-β-ᴅ-glucoside phlorizin (Fig. 1) was used as reference and is also oxidized by the enzyme (see Table S1 for summary). Earlier work with GlycDH applied O$_2$ as the oxidant. We show that the enzyme is ≥10$^2$-fold more active with K$_3$[Fe(CN)$_6$] (Table S1) and therefore use it instead of O$_2$ in the enzyme assay and in all synthetic reactions. In the course of this study, three other flavoenzymes were reported to perform C3 oxidation of *C*-glycosides[48–50]. These enzymes (ScPOx, CarA and *Ps*G3Ox) are distantly related to GlycDH (~20% identity) by common membership to the pyranose oxidase family (EC 1.1.3.10), but appear to be oxidases.

GlycDH reactions with phlorizin, puerarin and nothofagin are shown in Fig. 2a–d. HPLC and TLC were used to monitor the reaction progress (Figs. S6–S9) and the product structures were revealed by 1D- and 2D- NMR (Figs. S10–S16). GlycDH converts phlorizin cleanly into a single product (Figs. 2b, S6a, and S8), identified as 3-keto-phlorizin. The 3-keto-phlorizin is present partly (~41%) in the hydrated (gem-diol) form (Figs. S10 and S11). Kept on ice, the 3-keto-phlorizin (~2.0 mM; 50 mM HEPES buffer, pH 6.5) is sufficiently stable for further study of its enzymatic cleavage (see later).

GlycDH reaction with puerarin (Figs. 2c, S6b, and S8) gives a main product which based on NMR results (Figs. S12–S14) and

reactivity for glycosidic bond cleavage by *Pu*CGE (shown later) is identified as 3-keto-puerarin (Fig. 1). The minor species accounts for ~20% of the total product at all times (Fig. 2c) and is assigned as 2-keto-puerarin (Figs. S12–S14). Reducing agent (2.0 mM TCEP) is required to prevent the keto-products from decomposing during the reaction. The 3-keto-puerarin isomerizes spontaneously, as shown in earlier works[41], and the evidence here (Figs. 2c, S6b, and S17b) is consistent with equilibration of the 2- and 3-keto forms in solution faster than 3-keto-puerarin formation by the enzyme. The alternative interpretation, that GlycDH catalyzes a mixed C2/C3 oxidation of the puerarin, is less plausible, given the apparently fully equilibrated state of the 2- and 3-keto-products at each point of the reaction (Fig. 2c). GlycDH reaction with nothofagin likewise gives 2- and 3-keto-products (Figs. 2d, S6c, and S8). The product ratio depends on pH (Fig. S9a–c), with 3-keto preferred at pH 5.0 and 2-keto becoming gradually dominant as the pH is raised to 7.0. Two pieces of evidence demonstrate that enzymatic oxidation of nothofagin is selective at C3 and 2-keto product is formed only later in a spontaneous fashion. First, pH 6.5 reaction at 10-fold increased enzyme concentration reveals 3-keto-product formed much faster than 2-keto-product (Fig. S9d). Second, oxidized nothofagin comprised of ~90% 3-keto-product converts completely into the 2-keto-product in the absence of GlycDH (Fig. S9e) at a rate comparable to that observed in the enzymatic transformation (Fig. S9d).

## The GlycDH genomic context in *Agrobacterium tumefaciens* uncovers enzyme system of glycoside conversion, involving the β-eliminase *At*OGE

Lacking genome sequence of *Rhizobium* sp. GIN611, we applied BLAST protein search to the UniprotKB and NCBI databases focused on the *Rhizobium/Agrobacterium* taxonomic group to identify putative operons that involve a GlycDH homolog. Based on sequence similarity of ≥90%, numerous *Agrobacterium* sp., including the well-characterized *A. tumefaciens*, are found to encode a GlycDH in their genome. The genomic context of the GlycDH is highly conserved, as illustrated here on the example of *A. tumefaciens* FDAARGOS_1048 (strain identifier: CP066274.1). The GlycDH gene (WP_003515231.1) belongs to a putative operon (Fig. 3) that involves an additional oxidoreductase and two sugar phosphate isomerases/epimerases. Extending the search of GlycDH homologs, we find the putative operon organization to become increasingly diverse as the phylogenetic distance to the *Rhizobium/Agrobacterium* taxon increases (Fig. S18). Strikingly, therefore, the here discovered operon shows clear structural resemblance to the PUE strain operon for puerarin deglycosylation[41] as well as an operon of *Bacillus smithii S-2701M* for the degradation of levoglucosan (1,6-anhydro-β-glucopyranose)[51], as shown in Fig. 3. Regardless of difference at the individual enzyme level, the three operons appear to be analogous in the biochemical pathways encoded. Each pathway achieves, as predicted, a substrate deglycosylation that proceeds via oxidation, elimination, hydration and reduction (see Fig. S19 for the proposed levoglucosan pathway[51]). Given the focus of the current inquiry, our interest was naturally on the two *A. tumefaciens* putative sugar phosphate isomerases/epimerases (WP_003515232.1 and WP_038490972.1) to possibly catalyze glycosidic bond cleavage. It is interesting to note that genome search with the previously reported *C*-glycoside oxidases[48,49] did not reveal the particular operons identified with GlycDH.

Incubating the purified enzymes with 3-keto-phlorizin, we show with HPLC (Fig. S20) and TLC (Fig. S21) that putative sugar phosphate isomerase 1 converts the substrate under release of the phloretin aglycone. Results reported later in detail show 2-hydroxy-3-keto-D-glucal as the reaction product additionally, suggesting the enzyme as an *O*-glycoside β-eliminase, thus labeling it *At*OGE henceforth. Lacking

sequence similarity with CGEs or family GH4 glycoside hydrolases entirely, *At*OGE was a unique target to have its atomic structure determined. The putative sugar phosphate isomerase 2 is inactive towards 3-keto-phlorizin (Fig. S21). Preliminary evidence suggests its function in the hydration of 2-hyroxy-3-keto-D-glucal (labeling it *At*HYD henceforth), thus releasing 3-keto-D-glucose as substrate for reduction into D-glucose by the Gfo oxidoreductase from the *A. tumefaciens* operon (Fig. S22).

### The *At*OGE structure suggests convergent evolution of $Mn^{2+}$ active site for β-elimination of 3-keto-glycosides

The X-ray structure of *At*OGE was determined to 2.0 Å resolution using molecular replacement based on a RoseTTAFold[52] calculated ab initio model of the enzyme. A single *At*OGE dimer is present in the asymmetric unit. Each subunit adopts a TIM barrel fold, comprised of eight centrally arranged β-strands and seven exterior α-helices (Figs. 4a, S23, and S24 and Table S2). The dimer interface is formed by the α-helix α1 from each subunit (Fig. S23b). A 14-residue loop (Thr198−Gly211) between β-strand β7 and α-helix α7 as well as the C-terminal region (Leu236−Gln259; Fig. S24) show low electron density and are only partially modeled. A metal ion, likely $Mn^{2+}$ from the crystallization setup, is bound in a shallow cavity at the top entrance of the barrel. Despite the high similarity between the two chains in overall structure (root-mean-square deviation (RMSD): 0.221 Å), only one *At*OGE subunit (β) adopts the metal on the position that is plausible for substrate access. In the subunit β, it is coordinated by residues contributed from β-strands β5 (His132), β6 (Asp163), β7 (His189) as well as by a water molecule (Fig. 4b). Glu233 from β8 is weakly coordinating (Fig. 4b). The other subunit (α) shows the metal site shifted by ~5 Å and coordinated by His132, His66, Glu39 and Glu161 (Fig. S25). In this position, the metal is away from the highly conserved catalytic His134 (see later) and is in an unfavorable position for the proposed enzymatic reaction. Therefore, further analyses are made based on the active site as observed in subunit β.

The *At*OGE central fold is highly similar to that of *Pu*CGE$^α$ (RMSD: 1.230 Å) (Fig. 4a) as well as homologous subunits of other CGEs (Table S3 and Fig. S26)[45]. The metal site is also conserved, in regard to both the location within the β-barrel and the coordinating residues

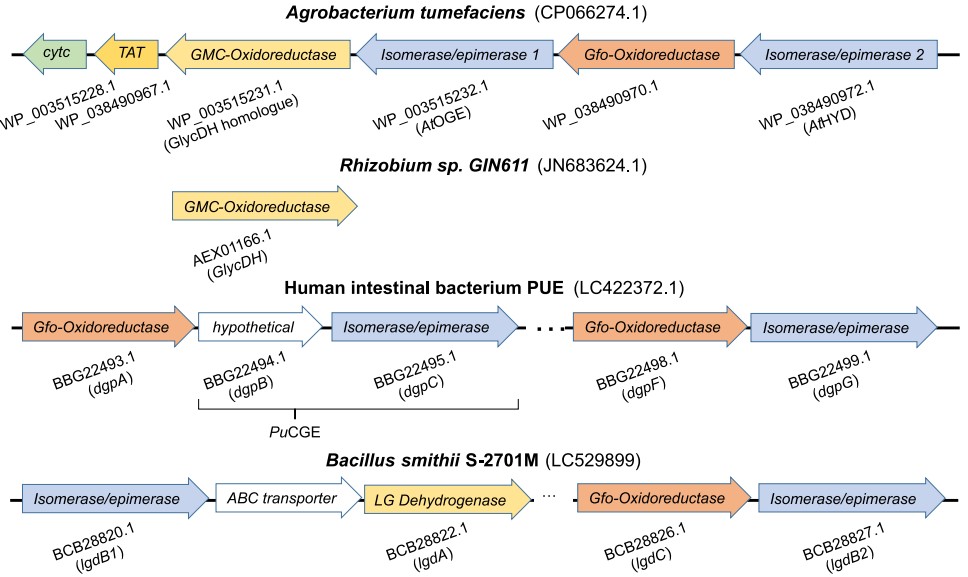

**Fig. 3 | *Agrobacterium tumefaciens* operon encoding a putative four-enzyme pathway of glycoside degradation.** The operon was identified due to the GlycDH homolog involved (95.5% sequence identity to the GlycDH from *Rhizobium* sp. GIN611). The *C*-glycoside degrading operon from the human intestinal bacterium PUE[41] and the levoglucosan degrading operon from *Bacillus smithii S-2701M*[51] have analogous structural organization and encode a comparable metabolic function. The predicted protein family, the accession number and the enzyme name abbreviation are given for each gene.

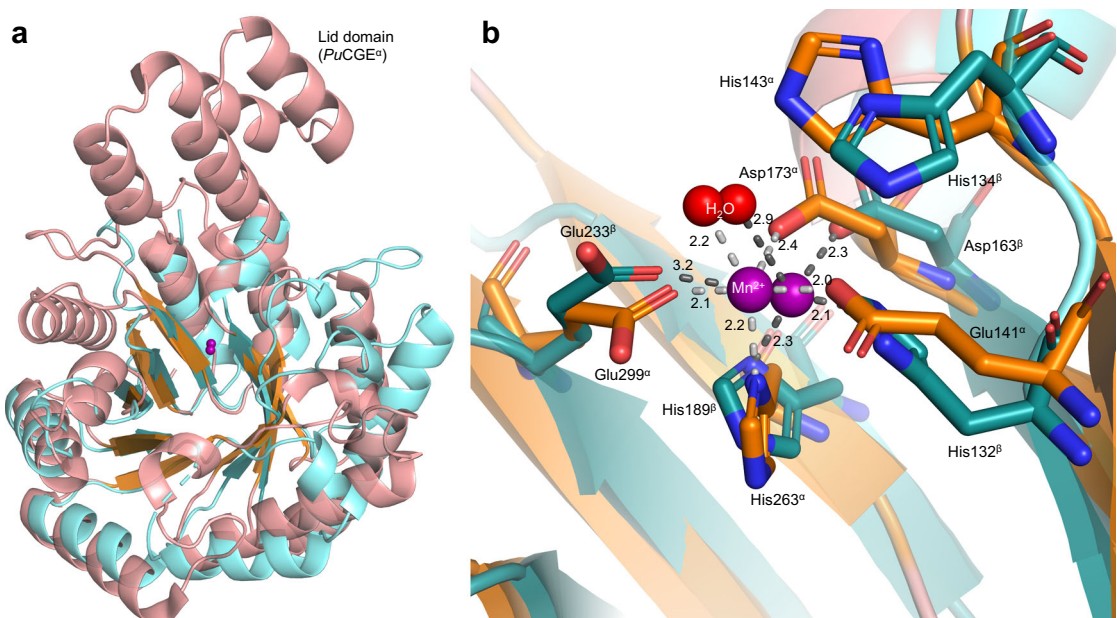

**Fig. 4 | Structural comparison of *At*OGE$^\beta$ to *Pu*CGE$^\alpha$. a** Superimposition of the central TIM barrel folds of *At*OGE$^\beta$ (cyan) and *Pu*CGE$^\alpha$ (salmon/orange), with the active site Mn$^{2+}$ shown as spheres. **b** Superimposition of the metal centers and the catalytic His of *At*OGE$^\beta$ and *Pu*CGE$^\alpha$. α = subunit A of *Pu*CGE; β = monomer B of *At*OGE.

used (Figs. 4a, b and S26). The coordinating His132 of *At*OGE is a glutamic acid in the CGEs (Figs. 4b and S26)[45]. *At*OGE lacks the additional four-helical lid domain (Figs. 4a and S24), that is a common, however not universally used structural element in the CGEs. Here the *At*OGE has closest resemblance to the short-form subunit of *Eubacterium cellulosolvens* CGE, as shown in Fig. S26. Combined effect of lid domain missing and second subunit (*Pu*CGE$^\beta$) absent makes the metal site of *At*OGE structurally wide open, ensuring broad substrate accessibility. The His residue implicated in CGE catalysis (His143, *Pu*CGE) according to literature[45,53] is conserved in *At*OGE (His134), as shown in Fig. 4b. Its functional importance in *At*OGE is shown by the corresponding H134A variant being completely devoid of activity (Fig. S27).

*At*OGE differs from the known CGEs in the extent of structural elements used to form an aglycone binding pocket. In the CGEs, the aglycone binding is very well developed and involves residues from both subunits[45]. Additionally, compared to *At*OGE, CGEs involve a more elaborate network of loops in the metal-binding region that contribute to shape the binding pocket (Fig. 4a, S24, and S26).

### Molecular dynamics simulations suggest substrate positioning for catalysis in *At*OGE and *Pu*CGE

We run extensive molecular dynamics (MD) simulations with explicit solvation in order to interrogate the binding mode of 3-keto-phlorizin and 3-keto-puerarin to the enzymes. Both substrates were placed at the active site by means of docking, and the selected docking poses were refined via MD simulations (see Methods for further details). The Michaelis complexes were stable under the conditions of simulation (Figs. S29–S33). In Fig. 5 (*At*OGE$^\beta$: 3-keto-phlorizin; Fig. 5a; *Pu*CGE: 3-keto-phlorizin, Fig. 5b, 3-keto-puerarin, Fig. 5c, and superposition of all three; Fig. 5d) can be seen that the 3'-keto-group in the substrates coordinates the Mn$^{2+}$ center whereas the 2'-hydroxy group of the glycosyl moiety interacts with a protein residue via hydrogen bonding (Fig. 5; Asp163 in *At*OGE$^\beta$; Glu301 in both *Pu*CGE complexes). In such a disposition, the conserved His residues in *At*OGE (His134) and *Pu*CGE (His143) are sterically well-suited and electronically activated for C2 proton abstraction (Fig. 5a–c; activated by Asn52b in *At*OGE$^\beta$; Asn144 in both *Pu*CGE complexes). In all enzyme complexes (Fig. 5), the His is furthermore close to the reactive oxygen or carbon of the leaving

group. The main differences in the binding of 3-keto-phlorizin in the two enzymes are found around the phloretin leaving group (Figs. 5, S35, and S36). Regarding 3-keto-puerarin bound to *Pu*CGE (Fig. 5c), the 7-hydroxyl of the daidzein leaving group is ionized (p$K_a = {\sim}6.5$) and stabilized by Tyr39$^\beta$ and Tyr303. The deprotonation of the *ortho* 7-hydroxyl to the reactive daidzein C6 and its stabilization are central to the proposed enzymatic mechanism of C-C bond fission (see Discussion). Importantly, some of the key interactions identified in MD simulations for *Pu*CGE in complex with 3-keto-phlorizin and 3-keto-puerarin and for *At*OGE in complex with 3-keto-phlorizin were validated experimentally via mutagenesis as shown later. We also monitor the puckering of the sugar moiety in 3-keto-puerarin and 3-keto-phlorizin when bound to the enzymes and in solution (Fig. S30). Whereas both substrates populate in major extension a $^4C_1$ conformation in solution, the binding to the metal center promotes conformations around $^4H_5/^5E$ for both substrates. Other minor conformations are also observed in 3-keto-phlorizin when bound to *Pu*CGE. Overall, the *At*OGE structure and its comparison with CGE (Figs. 4 and 5) suggest convergent evolution of Mn$^{2+}$ enzyme active site for 3-keto-glycoside deglycosylation via β-elimination.

### Biochemical comparison of *At*OGE and *Pu*CGE reveals distinct types of β-eliminase activity

To evaluate the two enzymes for *C*- or *O*-glycoside deglycosylation, we employed the product (3-keto-phlorizin) or the product mixture (2/3-keto-puerarin, 2/3-keto-nothofagin) as received from GlycDH glycoside oxidation. The already known reaction of *Pu*CGE with 3-keto-puerarin served as reference[41]. *Pu*CGE converts 3-keto-puerarin and 3-keto-nothofagin under release of the corresponding aglycone, daidzein and phloretin (Fig. 6a, b), with a specific activity of 18.6 (±0.41) U/mg and 0.64 (±0.06) U/mg, respectively (Table S4 and Fig. S37a). The $k_{cat}$ and the $K_m$ for the 3-keto-puerarin reaction were determined as 18 (±0.4) s$^{-1}$ and 0.8 (±0.1) mM, respectively (Fig. S37b and Table S4). Whereas 2-keto-nothofagin is not used by *Pu*CGE (Fig. S37c), the 2-keto-puerarin is converted in parallel with 3-keto-puerarin (Fig. S37d). MD simulations suggest that 2-keto-puerarin cannot be accommodated in a plausible position for reaction in the *Pu*CGE active site. Therefore, the 2-keto-puerarin

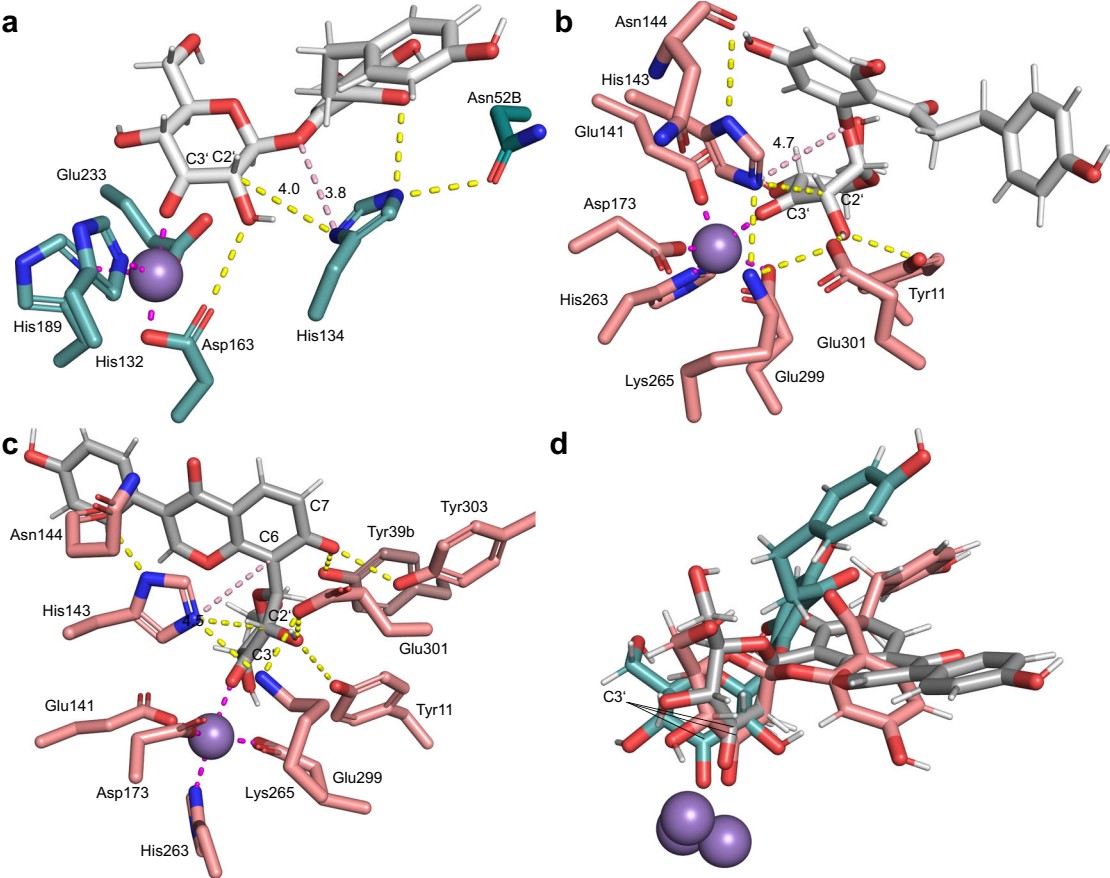

**Fig. 5 | Active-site close-up structures of *At*OGE and *Pu*CGE bound with 3-keto-glucoside substrate. a** *At*OGE and 3-keto-phlorizin, **b** *Pu*CGE and 3-keto-phlorizin, and **c** *Pu*CGE and 3-keto-puerarin. **d** Model **a**–**c** superimposed, showing Mn²⁺ coordinated substrate position. For each model, a representative structure was selected from the most populated cluster of conformers out of the three independent MD simulations (0.9 μs total time). Clustering was achieved using the hierarchical agglomerative (bottom-up) algorithm. Dashed lines show interactions: hydrogen bonds (distance <3.5 Å) in yellow; interactions with Mn²⁺ (distance <2.4 Å) in magenta; and distance (in Å) between the catalytic base (*At*OGE: H134; *Pu*CGE: H143) and the glycosidic oxygen or carbon in light pink.

conversion is likely apparent and reflects fast spontaneous 2,3-iso-merization under the conditions used (Figs. 6a and S37d). *At*OGE is inactive with both 3-keto-puerarin and 3-keto-nothofagin above a detection limit of ~0.01% of the *Pu*CGE activity (Fig. S38a). We further assessed *At*OGE in coupled reactions with GlycDH using orientin, vitexin and isovitexin (Fig. S7) as the substrates. While the GlycDH oxidizes each *C*-β-D-glucoside as expected, *At*OGE does not catalyze deglycosylation of the emerging keto-product.

Using 3-keto-plorizin, we show reaction of *At*OGE to release the phloretin aglycone (Fig. 6c) at a high specific activity, determined as 235 U/mg (±5) U/mg. Initial rate analysis (Fig. S39a) yields a $k_{cat}$ of 12 (±0.3) × 10² s⁻¹ and $K_m$ of 0.10 (±0.01) mM. *At*OGE is inactive towards the non-oxidized *O*-glycoside (phlorizin) (Fig. S38b). Interestingly, *Pu*CGE is also active with 3-keto-phlorizin, giving the same phloretin release from substrate consumption as shown in the *At*OGE reaction (Fig. 6d). The *Pu*CGE specific activity is lower considerably (~188-fold; 1.25 U/mg) than that of *At*OGE (Fig. S39), however it is comparable to the *Pu*CGE specific activity with 3-keto-nothofagin (Table S4). Kinetic parameters of *Pu*CGE with 3-keto-phlorizin were determined as $k_{cat}$ of 1.20 (±0.05) s⁻¹ and $K_m$ of 0.6 (±0.1) mM (Fig. S39b and Table S4).

Both *At*OGE and *Pu*CGE require metal ion (Mn²⁺) for 3-keto-glucoside cleavage (Fig. S40). Enzyme preincubation with EDTA (1.0–10 mM; 120 min) results in almost complete loss (≥90%) of the activity. Enzyme preincubated with Mn²⁺ (5.0 mM; 120 min) benefits the activity of *Pu*CGE (~11-fold). The *At*OGE activity is factually unaffected by external Mn²⁺, suggesting that the isolated *At*OGE involves a fully loaded metal center. Structural perturbation of the Mn²⁺

metallocenter in a H189A variant of *At*OGE disrupts the activity for 3-keto-phlorizin cleavage to below 0.1% wildtype level (Fig. S27). The result agrees with Mori et al.[45] who reported the homologous His → Ala substitution in *Pu*CGE.

While aglycone release is monitored conveniently by HPLC with reference to authentic standards, the glycone-derived product of the enzymatic reactions required direct identification. The *At*OGE product from incubation with C3 oxidized phlorizin is shown as 2-hydroxy-3-keto-D-glucal (1,5-anhydro-D-*erythro*-hex-1-en-3-ulose) based on a comprehensive set of 1D (Figs. S41 and S42), 2D (Fig. S43) and in situ NMR data (Fig. S44). The same product is formed in *Pu*CGE reactions with the 3-keto-*C*-β-D-glucosides (Fig. S45), as expected from literature on the enzyme's reaction with 3-keto-puerarin[44,53]. And importantly, it is also the product of the *Pu*CGE-catalyzed *O*-glycoside deglycosylation of 3-keto-phlorizin (Fig. S46).

In summary, these results identify *At*OGE as a Mn²⁺-dependent, *O*-glycoside-specific OGE that shows high specific activity with 3-keto-phlorizin. *Pu*CGE is a promiscuous CGE-OGE that exhibits lower activity than *At*OGE in *O*-glycoside deglycosylation. All enzymatic reactions yield the same 3-keto-D-glucal elimination product (Fig. 6e).

### Stereochemical preference and rate acceleration of *At*OGE in 3-keto-glucoside cleavage

Anomeric specificity in the cleavage of 3-keto-glucoside substrates represents an important characteristic of OGE catalytic function that can be mechanistically informative. We synthesized two pairs of α/β-configured 3-keto-glucosides (Figs. S47–S50), differing in intrinsic

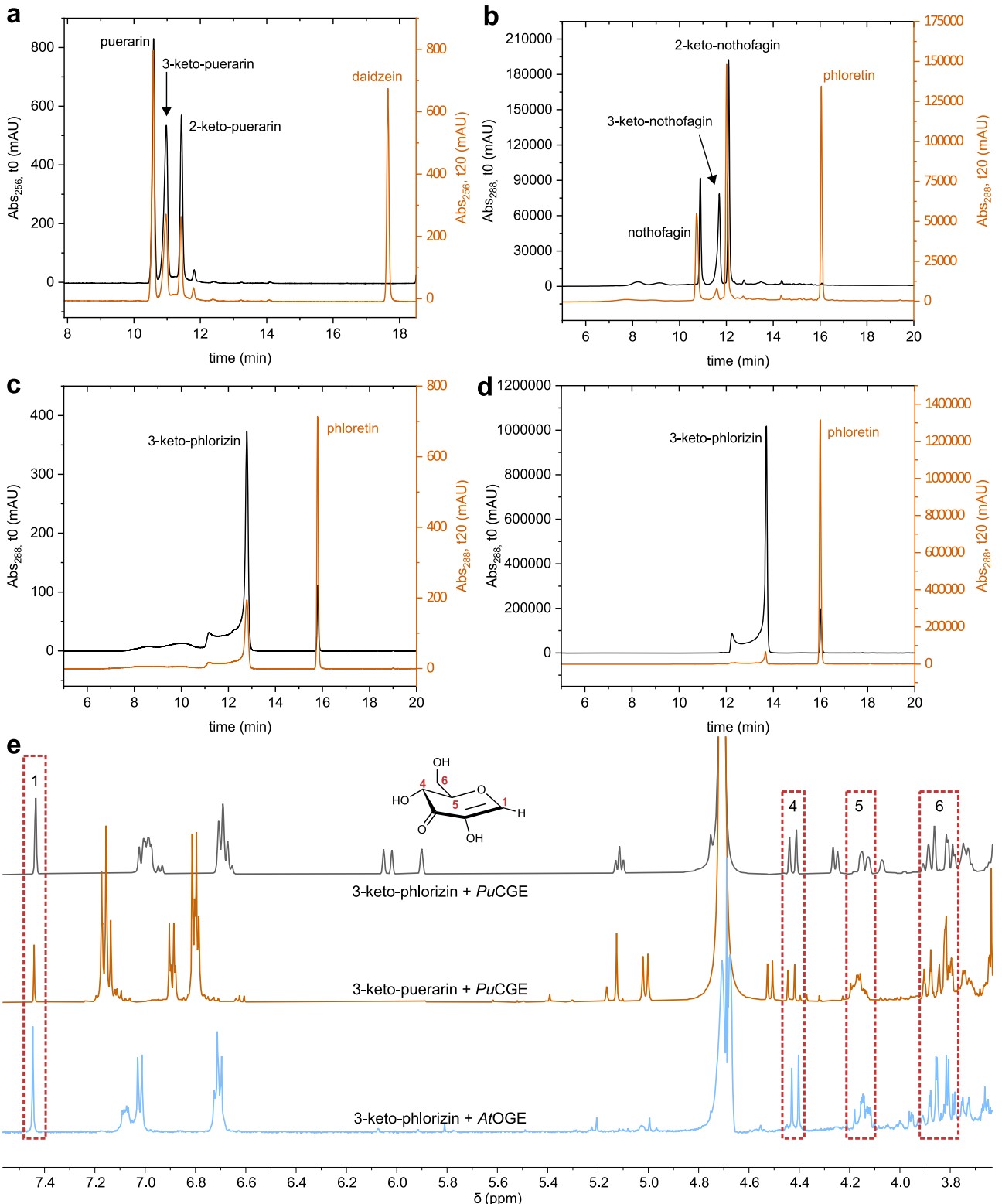

**Fig. 6 | Products released in enzymatic deglycosylation of 3-keto-*O*/*C*-gluco-sides. a–d** HPLC chromatograms showing *Pu*CGE (**a–c**) and *At*OGE reaction (**d**), with absorbance traces at zero time and 20 min shown in black and orange, respectively. Conditions: 10 mM potassium phosphate buffer (pH 6.0), 37 °C, 2.0 mM substrate. *Pu*CGE (0.10 mg/mL in **a**, **c**, and 1.00 mg/mL in **b**), *At*OGE (0.01 mg/mL in **d**). **e** ¹H NMR spectra of reaction mixtures (500 MHz, D₂O). Signals of 2-hydroxy-3-keto-ᴅ-glucal are framed in red (*n* = 1 individual experiment).

reactivity of the leaving group as expressed by the p$K_a$ (4-nitrophenol: 7.0; glucose: ~12 for the 4-OH), and used them as substrates of *At*OGE. Based on leaving group released (Fig. S51a–c), the enzyme is active with both diastereomers of 4-nitrophenyl 3-keto-glucoside (note: ᴅ-*arabino*-hexopyranosid-3-ulose), preferring the β- (39 U/mg) over the α-configuration (1.2 U/mg); and it exhibits high activity with the β-configured keto-disaccharide (3′-keto-cellobiose, β-ᴅ-*ribo*-hexopyr-anosyl-3-ulose-(1→4)-ᴅ-glucopyranoside) whereas it does not cleave

the α-configured 3′-keto-maltose (α-D-*ribo*-hexopyranosyl-3-ulose-(1→4)-D-glucopyranoside), even at extended reaction times (20 h). NMR data identify 2-hydroxy-3-keto-D-glucal released as the second product in all reactions giving glycoside cleavage (Figs. S52 and S53). Interestingly, *Pu*CGE is inactive with both 3′-keto-disaccharides (Fig. S51d).

The *At*OGE kinetic parameters for conversion of 3′-keto-cellobiose are $k_{cat}$ of 51 (±4) $s^{-1}$ and $K_m$ of 2.6 (±0.1) mM (Fig. S51e). In terms of catalytic efficiency ($k_{cat}/K_m$), the enzyme prefers 3-keto-phlorizin ($1.2 \times 10^6 M^{-1}s^{-1}$) about 61-fold over 3′-keto-cellobiose ($6.1 \times 10^4 M^{-1}s^{-1}$). In a first effort at quantifying the rate acceleration provided by *At*OGE, we determined the apparent first-order rate constant ($k_{non}$) for spontaneous cleavage of 3-keto-phlorizin, which is ~8.6 (±1.5) $\times 10^{-6} s^{-1}$ (Figs. S17a and S54). From the ratio $k_{cat}/k_{non}$, thus, an estimate of $1.4 \times 10^7$ is obtained.

### Mutagenesis of *Pu*CGE shows interactions essential for the *C*-glycoside bond cleavage

To identify elements of the protein structure unique to *Pu*CGE and critical for *C*-glycoside deglycosylation, we pursued a two-pronged approach. Overall goal was to interfere with substrate positioning in the aglycone binding pocket as formed by the β subunit (see Fig. 5c). One strategy involved complete disruption of the relevant part of the binding pocket by deleting the β subunit entirely. The other was site-specific and probed the oxyanion binding pocket interactions of Tyr39β from the β subunit (see Fig. 5c). Of note, Tyr39β is a conserved residue in the known CGEs (Figs. S55 and S56). Tyr39β was previously recognized as part of the enzyme's aglycone binding pocket[45], however, a possible role of the residue in the immediate *Pu*CGE reaction was not hitherto anticipated. The isolated *Pu*CGEα subunit and the *Pu*CGE dimer with Tyr39β substituted by Phe were assessed in reactions with 3-keto-phlorizin, 3-keto-puerarin and 3-keto-nothofagin. The *Pu*CGEα subunit retains activity at a low level (~1.4% of the native *Pu*CGE) with 3-keto-phlorizin while it is completely inactive with the 3-keto-*C*-β-glucosides (Fig. S57 and Table S5). The mutated *Pu*CGE exhibits only a minor loss of 3-keto-phlorizin activity (2.4-fold; 59%) compared to the wild-type enzyme while it has lost effectively all of the activity (-1.2 × 10⁴-fold; >99%) towards cleavage of the 3-keto-*C*-β-glucosides (Fig. S57 and Table S5). These results implicate the importance of the β subunit in the formation of a fully functional *Pu*CGE enzyme. Most importantly, they demonstrate Tyr39β as specifically important to the CGE activity of *Pu*CGE.

### Family GH4 glycoside hydrolases do not cleave phloretin *C*-glucoside

Given the clear mechanistic analogy among OGE/CGE and family GH4 glycoside hydrolases (Figs. S2–S4) in the β-elimination step of glycoside bond cleavage, we were interested in clarifying whether the glycoside hydrolases use *C*-glycosides as substrates of deglycosylation. We chose two well-characterized GH4 enzymes (GlvA from *Bacillus subtilis*[54]; BglT from *Thermotoga maritima*[32]). Both enzymes require a 6-phospho hexopyranoside as substrate. They differ in preference for the substrate anomeric configuration (α for GlvA, β for BglT), yet can utilize α- and β-configured glucosides with good leaving groups[33]. The 6-phospho derivatives of phlorizin and nothofagin were synthesized (Figs. S58 and S59) and assessed for reactivity with GlvA and BglT, based on measurement of aglycone release. Note that phloretin is a relatively good leaving group, rendering both GlvA and BglT as interesting candidates to be evaluated. Both enzymes cleave 6-phospho-phlorizin with low activity, yet they do not cleave 6-phospho-nothofagin (Figs. S60–S63). Kinetic parameters for the 6-phospho-phlorizin are (Figs. S60d and S61d and Table S4): GlvA, $k_{cat} = \sim 0.02 \, s^{-1}$, $K_m = \sim 0.1$ mM; BglT, $k_{cat} = \sim 0.01 \, s^{-1}$, $K_m = \sim 0.3$ mM.

## Discussion

Four-enzyme pathway for the degradation (formally a net hydrolysis; Fig. 7a) of natural product *O*-β-glucosides is discovered in *Agrobacterium tumefaciens*. The pathway is widespread in the *Rhizobium/Agrobacterium* taxon as well as in numerous other groups of plant-associated bacteria. The different steps of the pathway (Fig. 7a), that is, glucoside C3 oxidation, β-elimination for glycosidic bond cleavage, 1,4-Michael addition of water and reduction, are even more broadly conserved among bacteria, based on enzymes that are unrelated by sequence but retain the same biochemical function[44,51]. Family GH4 glycoside hydrolases are truly astonishing for their ability to accommodate the entire reaction pathway within a single active site[30–38,55]. Given the broad specificity of the flavoenzyme catalyzing glycoside C3 oxidation in the first step (see Kim et al.[46,47] and results of this study), the biochemical scope of the *A. tumefaciens* pathway appears to be effectively determined by *At*OGE, a previously uncharacterized Mn²⁺ metalloenzyme here identified as OGE. From its structure and function, *At*OGE is suggested to represent an archetype of β-eliminase for the deglycosylation of natural product 3-keto-*O*-β-glucosides. The appearance of effectively the same/highly similar Mn²⁺ active site in CGE and family GH4 glycoside hydrolase for a largely analogous catalysis of β-elimination suggests the agency of convergent evolution in these enzymes, with CGE[41,45] and family GH4 glycoside hydrolase[30,31,55] representing gain in function compared to *At*OGE.

Mechanistic proposal for *At*OGE (Fig. 7b) involves Mn²⁺ in substrate positioning and catalysis. The Mn²⁺ would assist in the proton abstraction by His134 through polarization of the carbonyl group as well as stabilization of the emerging enolate. Deprotonation at C2 would be further facilitated by hydrogen bonding from the C2-OH to a protein residue. Expulsion of the leaving group warrants more research to clarify its relative timing in the overall β-elimination as well as its reliance on general acid catalytic support. Evidence for GH4 glycoside hydrolases shows that in the conversion of aryl glycoside substrates, departure of the leaving group requires no protonic assistance or is only weakly catalyzed[33,35–38], and both chemical and non-chemical steps may be rate-limiting in this process[37]. We consider here a tentative scenario for *At*OGE in which His134 (in the conjugate acid form) might also provide proton catalysis to the expulsion of a poor leaving group of high p$K_a$. The *syn* stereochemical course of the elimination reaction thus implied would explain the requirement for a β-glucoside structure of substrate when a poor leaving group, such as the C4 alcohol of glucose, is involved. An activated leaving group, such as 4-nitrophenol or phloretin (Fig. 7b), can depart without assistance from His134, perhaps not even hydrogen bonding[37], as would also be the case for the α-configured 3-keto-glucoside substrate. However, specificity for reaction with the β-glucoside remains, as expected. Results of our extended MD simulations show that the δN of the imidazole in His134 stays close to both sugar H2 and ether oxygen of the leaving group (Fig. S32).

Based on the collective evidence obtained in this study, enzymatic cleavage of 3-keto-*C*-β-glucosides is understood as an extension of the basic *At*OGE mechanism of β-elimination, distinct from Mori et al.[47] To proceed, the reaction requires specific actuation by enzyme-aglycone interactions that in natural CGEs appear to be mediated partly from an additional protein subunit. The *Pu*CGE complex with 3-keto-puerarin elucidated in the MD simulations (Fig. 5c) shows Tyr39β together with Tyr303 forming an oxyanion binding pocket, well-suited to accommodate the relevant daidzein 7-OH (*ortho* to the reactive C6) in an ionized form. Given the low p$K_a$ of ~6.5 for the 7-OH, *Pu*CGE may bind the substrate oxyanion from solution. Alternatively, Tyr39β could act as base for 7-OH deprotonation. All known CGEs have the Tyr39β homolog engaged in a hydrogen bonding/electrostatic network of structural interactions, apparently suitable for modulation of the side chain p$K_a$ of the tyrosine (Fig. S58). As we show in Fig. 7c, when the

**Fig. 7 | Proposed *O*- and *C*-glycoside deglycosylation reactions.** Pathway of phlorizin degradation in *A. tumefaciens* (**a**) and mechanisms for the elimination reactions catalyzed by OGE (**b**) and CGE (**c**). In (**b**), the protein residues shown are for *At*OGE. The OGE reaction in (**b**) is shown as a rather stepwise process,

considering that due to the expected similarity in the p$K_a$ values of His134 and phloretin leaving group the C-O bond cleavage would have to be far advanced before proton transfer could occur. With other substrates involving more basic leaving group, the two steps may show higher degree of concertedness.

*ortho* hydroxy group is deprotonated, carbanion character can be generated at the reactive carbon simply by resonance (without requiring an energy-intensive dearomatization)[45], thus making it a suitable leaving group for the elimination under protonic assistance from the catalytic histidine (His143 in *Pu*CGE). The proposed mechanism implies *C*-glycoside deglycosylation restricted to aglycone structures possessing *ortho* or *para* hydroxy groups, hence are amenable for carbanion induction by resonance. The known CGE substrates all fulfil the structural requirements of the proposed mechanistic scenario[45,53]. Regarding the activation of the polyphenol group for reaction at carbon as opposed to reaction at oxygen, there exist clear mechanistic analogies between OGE/CGEs and *O/C*-glycosyltransferases[9] as well as *O/C*-alkyltransferases (e.g., methyl, prenyl).[56–58] In all cases, enzymatic reactivity at carbon is thought to arise from an extension of the basic mechanism for the reaction at oxygen, and it appears to require precise positioning together with reactive carbon site activation by resonance as a synergistic combination of essential catalytic factors (Fig. S64).

Lastly, our study has importance in suggesting a biological role for the two-step enzymatic C3 oxidation–elimination in the microbial utilization of plant-derived natural product *O*- and *C*-glycosides. The discovery that CGE exhibits dual specificity, with both 3-keto-*O*- and *C*-glycosides used as substrates, could suggest a generalist function of the enzyme in glycoside cleavage. The versatile catalysis of CGE appears to be consistent with the enzyme's widespread natural distribution in different microbiota[45] that feed on plant material (e.g., the human microbiome) or involve commensalism with plants. After all, plant natural product glycosides mostly involve an *O*-glycosidic structure and the *C*-glycosides typically constitute only a minor portion of the total[59,60]. Additionally, the *O*-glycosides embody a much greater diversity of aglycone structures than the *C*-glycosides[61]. The sequence of C3 oxidation and elimination is suggested as a powerful strategy of biological catalysis to achieve *O*-glycoside deglycosylation with limited reliance on the variable aglycone for rate enhancement and specificity. The challenging task of *O*-glycoside cleavage is thus subdivided into two steps, with the effect that the enzymatic rate acceleration required in the immediate glycosidic bond fission is decreased by several orders of magnitude compared to the reaction of a glycoside hydrolase acting on the corresponding non-oxidized glycoside substrate[24,29]. The physiological function of natural product *O*-glycoside deglycosylation with relaxed aglycone specificity might be more effectively managed by a β-eliminase working in pair with a C3 glycoside oxidoreductase than by an aglycone-promiscuous glycoside hydrolase (e.g., see Theilmann et al.[28] and Németh et al.[18]). The $10^7$-fold rate acceleration by *At*OGE in the cleavage of 3-keto-phlorizin can provide distinct advantage over spontaneous reaction proposed by other authors[47]. The molecular distinction between *At*OGE- and *Pu*CGE-type β-eliminases revealed in the current study enables precision in genome screening for *O*- compared to *C*-glycoside deglycosylation activity. The suggested involvement of GH4- and GH109-type glycoside hydrolases[30,40,62] in the degradation of neutral oligosaccharides and glycans emphasizes the versatile biological uses of oxidation-driven enzymatic β-elimination for glycoside cleavage.

## Methods

### Chemicals, enzymes, and microbial strains

Unless stated otherwise, all chemicals were from Carl Roth (Karlsruhe, Germany), Sigma Aldrich (St. Louis, MO, U.S.A.) or Merck KGaA (Darmstadt, Germany) at their highest available purity. The glycosides and flavonoid aglycones were from Carbosynth Ltd. (Compton, UK), except for phlorizin (Carl Roth) and phloretin (AK Scientific, Inc., Union City, NJ, U.S.A.). The expression strain *E. coli* Lemo21 (DE3) was from New England Biolabs GmbH (Frankfurt/Main, Germany). Synthetic genes were codon-optimized and provided by GenScript (Piscataway Township, NJ, U.S.A.). SDS-PAGE was performed in the XCell

SureLock Mini-Cell Electrophoresis System with ready-to-use NuPA-GETM 4.0–12% Bis-Tris 1.0 gels (Invitrogen by Thermo Fisher Scientific, Inc., Waltham, MA, U.S.A.). Hexokinase assay was from Human Diagnostics Worldwide (Wiesbaden, Germany).

### Phylogenetic analysis of putative glycoside degrading operons

Using the protein sequence of GlycDH from *Rhizobium sp. GIN611* (Accession number: AEX01166.1) as a query, a BLAST search was performed against the UniProtKB protein database within prokaryotic organisms. Using the NCBI Reference Sequences, the genomic context of hits with an identity of >70% was investigated and compared with each other as well as with the putative puerarin metabolizing operon PUE. Genes in close vicinity to the GlycDH homolog were inspected based on their predicted protein family. Microorganisms with similar operon arrangements were subject to phylogenetic comparison.

### Protein expression and Strep-trap purification

Expressed enzymes and their herein used abbreviations in bold (GenBank/NCBI reference sequence in brackets) were the following: Oxidoreductase **GlycDH** (AEX01166.1) and TAT-pathway signal protein **TAT** (AEX01167.1) from *Rhizobium sp. GIN611*; 6-phospho-β-glucoside kinase **BglK** (AAK58463.1) from *Klebsiella pneumoniae*; *C*-glycoside eliminase ***Pu*CGE** (BBG22494.1 and BBG22495.1) and variant *Pu*CGE-Y39F from PUE strain; *O*-glycoside eliminase ***At*OGE** (WP_003515232.1), variant *At*OGE-H134A, variant *At*OGE-H189A, putative hydratase ***At*HYD** (WP_038490972.1) and putative oxidoreductase **Gfo Oxo** (WP_038490970.1) from *Agrobacterium tumefaciens*; maltose-6'-phosphate glucosidase **GlvA** (WP_088325462.1) from *Bacillus subtilis*; 6-phospho-beta-glucosidase **BglT** (WP_004079953.1) from *Thermotoga maritima*. Gene sequences are summarized in Table S6. The expression vectors (pET-Duet-1_*glycdh*_tat, pET17b-*bglk*, pET17b-*bglt*, pET-Duet-1 *pucge*α *pucge*β, pET-Duet-1 *pucge*α-*Y39F*_ *pucge*β, pET17b-*atoge1*, pET17b-*atoge-H134A*, pET17b-*atoge-H189A*, pET17b-*athyd*, pET17b-*oxo* and pET17b-*glvA*) were obtained from GenScript and transformed into the *E. coli* Lemo21 (DE3) expression strain by electroporation. The expression plasmid encoding the *Pu*CGEα subunit was produced by overlap extension PCR using primers shown in Table S7 and pET-Duet-1 *pucge*α *pucge*β as a template (see the subsection, Cloning of pET-Duet-1 *pucge*α). Transformants were selected over 16 h at 37 °C on LB-agar plates containing antibiotics (100 μg/mL of ampicillin for all expression strains, 25.0 μg/mL of kanamycin and 2.50 μg/mL of tetracycline additionally for GlycDH and TAT expression).

All enzymes were designed to have an *N*-terminal Strep-tag II. They were produced using a standard protocol. Cultivation was performed in baffled 1-L shaking flasks using 250 mL of selective LB-media (100 μg/mL of ampicillin, except for GlycDH and TAT expression 50.0 μg/mL ampicillin, 12.5 μg/mL of kanamycin and 1.25 μg/mL of tetracycline) at 37 °C and 110 rpm in a Certomat BS-1 shaker from Sartorius (Göttingen, Deutschland). The medium was inoculated to an OD600 of 0.1 and at an OD600 of 0.9, gene expression was induced by 0.1 mM IPTG (isopropyl β-D-1-thiogalactopyranoside) at 18 °C for 20 h. GlycDH/TAT expression was induced at a lower OD600 of 0.4–0.6. Cells were harvested by centrifugation at 1930 × *g* and 4.0 °C for 10 min with a HiCen SR refrigerated high-speed centrifuge (Herolab, Wiesloch, Germany). The supernatant was discarded and the pellet resuspended in 30 mL of loading buffer (100 mM Tris/HCl, pH 8.0 with 150 mM NaCl and 1.0 mM EDTA) and a spatula tip of lysozyme (Carl Roth). The cells were disrupted on ice for 5.0 min using a Fisherbrand™ Model 505 Sonic Dismembrator (Fisher Scientific) and the cell free extract was separated through centrifugation at 4.0 °C and 16,000 × *g* for 45 min with an Eppendorf 5415 R micro centrifuge (Vienna, Austria). The lysate was treated with a 0.2 μm Sartorius syringe filter and loaded onto two 5.0 mL StrepTrap™ HP columns (GE Healthcare, Little Chalfont, UK) equilibrated in loading buffer. The columns were installed on an ÄKTA prime plus chromatography system from GE Healthcare and

unspecific protein was washed out with same buffer. The Strep-tagged protein was eluted with loading buffer, supplemented with 2.5 mM desthiobiotin using a flow rate of 5.0 mL/min at 7.0 °C. The fractions containing the protein of interest were pooled, concentrated and buffer exchanged in a Sartorius Vivaspin™ Turbo 15 via centrifugation at $5975 \times g$ and 4.0 °C with an Eppendorf Centrifuge 5810R.

The enzymes were stored in 50 mM 4-(2-hydroxyethyl) piperazine-1-ethanesulfonic acid (HEPES) buffer (pH 7.0) at concentrations between 10.0 and 50.0 mg/mL at −20 °C until further usage. GlycDH/TAT was supplemented with 50 μM flavin adenine dinucleotide). Additionally, for BglK 250 mM NaCl and 5.0% (v/v) glycerol were added for proper storage. The purification process was monitored via a 280 nm UV-detector and SDS-PAGE. The PAGE analysis is displayed in Fig. S66.

### Synthesis and isolation of keto-glycosides

Synthesis was performed at 2.0–20 mL scale, with 2.0–20 mM of the glycoside substrate being dissolved in 100 mM potassium phosphate buffer (pH 5.7). To reactions with the *C*-glycosides or with phlorizin, 10% (v/v) of dimethyl sulfoxide (DMSO) was added as cosolvent. The buffer contained 8.0–50 mM of potassium ferricyanide ($K_3[Fe(CN)_6]$) for all reactions as indicated, and 2.0 mM of tris-(2-carboxyethyl)-phosphine (TCEP) additionally for the reactions with puerarin. Reactions were initiated by GlycDH addition and the concentration was adjusted dependent on the substrate used: maltose, cellobiose, phlorizin (0.02 mg/mL), puerarin, vitexin, isovitexin, orientin (0.25 mg/mL), nothofagin (0.10 mg/mL), 4NPαG and 4NPβG (both 0.04 mg/mL). The reactions were incubated at 650 rpm in a Thermomixer (Eppendorf) when reaction volumes were 2.0 mL, or at 120 rpm in a Certomat BS-1 shaker (Sartorius), when above, either way at 37 °C. The product formation was followed by HPLC, TLC or photometrically. Upon completion, the reactions were quenched by adding double the volume of a 4:1 mixture of MeCN and EtOAc. The two-phase mixture was then shaken on a Stuart SB3 rotator (Stuart Equipment, Stone, UK) for 15 min and left at 4.0 °C for an additional 45 min. The organic phase, which contained the keto-glycoside, was collected and concentrated using reduced pressure (20 mbar, 40 °C, 100 rpm) in a Laborota 4000 efficient rotary evaporator (Heidolph Instruments, Schwabach, Germany) until the volume was between 0.5 and 2.0 mL. The concentrated solution was then subjected to shock freezing using liquid $N_2$ and lyophilization was performed for 16 h using a Christ Alpha 1-4 freeze drier (bbi-biotech GmbH, Berlin, Germany) with a Vacuubrand pump unit RZ 6.

GlycDH reactions with disaccharides were treated differently. Following completion of the reaction, enzyme was filtered off using Amicon Ultra-15 Centrifugal Filter Unit at $5975 \times g$ and 4.0 °C with an Eppendorf Centrifuge 5810 R. The $K_3[Fe(CN)_6]$ was removed by incubation with Amberlite® IRC120 H (DuPont, Wilmington, DE, U.S.A.). The aqueous preparation was concentrated, shock frozen and lyophilized in the same manner as the other keto-glycosides.

The purity and structures of the isolated compounds were confirmed using $^1H$ NMR (see NMR sections in Methods for protocol). The concentrations were measured photometrically (see respective section below).

### Preparation of 6-phospho-D-glucosides

Synthesis was performed at 15 mL scale in Sarstedt tubes (Biedermannsdorf, Germany). Reactions contained 5.0 mM substrate (phlorizin, nothofagin; note: puerarin was not accepted by BglK), 50.0 mM HEPES buffer (pH 6.5), 10% (v/v) DMSO, 1.0 mM MgCl$_2$, 7.5 mM ATP and 1.00 mg/mL BglK. Reactions were incubated at 37 °C and 120 rpm in a Certomat BS-1 shaker (Sartorius) for up to 120 min and were monitored by HPLC and TLC analysis. Following completion of reaction, BglK was removed by a Satorius Vivaspin™ Turbo 15 via centrifugation at $5975 \times g$ and 4.0 °C with an Eppendorf Centrifuge

5810 R. The permeate was loaded onto a Herka Interfit® 200 mm chromatography column (Herka Laborgeräte GmbH & Co. KG, Kreuzwertheim, Germany) packed with 60 Å silica gel from Merck KGaA. The compounds were separated using a 9:1 mixture of MeCN and $H_2O$. Elution of compounds was followed by TLC and using a DS-11+ spectrophotometer at the respective absorption maxima (288 nm for phosphorylated phlorizin and nothofagin,). The pooled fractions were concentrated under reduced pressure (20 mbar, 40 °C, 100 rpm) to a total volume of 2.0 mL using a Laborota 4000 rotary evaporator (Heidolph Instruments). The concentrated solution was then subjected to shock freezing using liqid nitrogen and lyophilization for 16 h using a Christ Alpha 1-4 freeze drier (bbi-biotech GmbH) attached to a Vacuubrand pump unit RZ 6. The lyophilizates of nothofagin6P and phlorizin6P were resuspended in 50 mM HEPES of pH 6.5 and pH 7.5, respectively. The concentration was about 1.0–2.0 mM and the mixtures were stored at −20 °C until further usage.

### Photometric determination of protein and glycoside concentrations

Absorbance measurements were performed using a DS-11 Spectrophotometer (DeNovix, Wilmington, DE, U.S.A.). Measurements were performed at the respective absorption maxima and concentrations were estimated based on the molar extinction coefficient: 3-keto-phlorizin, 3-keto-nothofagin, phlorizin6P and nothofagin6P (288 nm, 13 AU mM$^{-1}$ cm$^{-1}$); 3-keto-puerarin (256 nm, 33 AU mM$^{-1}$ cm$^{-1}$); 4NPα3-ketoG(6P) and 4NPβ3ketoG(6P) (310 nm, 9.3 AU mM$^{-1}$ cm$^{-1}$); 3'-keto-maltose (340 nm, 5.50 AU mM$^{-1}$ cm$^{-1}$)[63] and 3'-keto-cellobiose (340 nm, 3.55 AU mM$^{-1}$ cm$^{-1}$)[63].

Protein concentrations were measured at 280 nm and concentrations were calculated using the corresponding molar extinction coefficient and the molecular weight computed by ExPASy ProtParam tool[64]. GlycDH (66,308 Da, 118,635 M$^{-1}$ cm$^{-1}$), BglK (34,919 Da, 42,650 M$^{-1}$ cm$^{-1}$), *At*OGE (29,895 Da, 33,460 M$^{-1}$ cm$^{-1}$), GlvA (51,960 Da, 68,090 M$^{-1}$ cm$^{-1}$), BglT (48,997 Da, 40,925 M$^{-1}$ cm$^{-1}$), *At*HYD (41,023 Da, 67,630 M$^{-1}$ cm$^{-1}$), *Pu*CGEα (39,205 Da, 51,270 M$^{-1}$ cm$^{-1}$), *Pu*CGEβ (18,313 Da, 28,545 M$^{-1}$ cm$^{-1}$), *Pu*CGE$^{(α+β)}$ (57,500 Da, 79,815 M$^{-1}$ cm$^{-1}$)

### Characterization of spontaneous isomerization of 3- to 2-keto-nothofagin

The reactions were performed in 200 μL scale and incubated at 37 °C and 650 rpm (Eppendorf Thermomixer). Standard conditions used were 100 mM potassium phosphate buffer (pH 6.5), 10% (v/v) DMSO, 8.0 mM $K_3[Fe(CN)_6]$, 2.0 mM nothofagin and GlycDH (0.10 mg/mL). To evaluate the effect of pH on the 2- and 3-keto-equilibrium, the reactions were performed at pH 5.0, 6.0 and 7.0. In a separate reaction, excess enzyme (1.00 mg/mL) was added to achieve oxidation at a rate faster than the 3- to 2-keto-isomerization rate. As a control, isomerization of 2.0 mM isolated 2/3-keto-nothofagin was monitored under standard conditions in the absence of GlycDH and $K_3[Fe(CN)_6]$. Samples (20 μL) were quenched with cooled MeCN (1:1, v:v) and analyzed on HPLC.

### Measurement of spontaneous deglycosylation of 3-keto-phlorizin

The reactions were conducted at 200 μL scale and incubated at 37 °C and 650 rpm (using an Eppendorf Thermomixer) for up to 360 min. 3-Keto-phlorizin was used at a concentration of 2.0 mM, 1.0 and 0.5 mM. Incubations were done in 50 mM HEPES buffer (pH 6.5). To serve as a positive control, a reaction was performed by adding *At*OGE ($5.00 \times 10^{-3}$ mg/mL; 0.17 μM). To mimic the conditions of the enzymatic reaction, MnCl$_2$ was added and histidine (the active-site residue of *At*OGE) was supplied in free form in indicated 1000-fold concentration to *At*OGE (167 μM). Samples (20 μL) were quenched with cooled MeCN (1:1, by volume) and analyzed on HPLC. The apparent first-order rate constant ($k$) was determined by global fit of time

courses recorded at varied substrate concentration [$S_0$] to Eq. 1, where $t$ is the time and [$S_t$] the substrate concentration at a given time. Fitting was done with SigmaPlot V10.0 (Systat, Erkrath, Germany).

$$[S_t] = [S_0]\, e^{-kt} \tag{1}$$

### Initial rate analysis to determine kinetic parameters

Incubations were done in a total volume of 200 µL using an Eppendorf thermomixer with agitation at 650 rpm at 37 °C. The detailed reaction conditions varied based on the enzyme used and are stated in the subsections below. To initiate the reactions, concentrated enzyme solution (≤2% of total volume) was added to the substrate solution after temperature equilibration. Activity determinations were performed under conditions of substrate saturation. For kinetic parameter determination, initial rates were measured at variable substrate concentrations (≥6 concentrations). At specified times, samples of 20 µL were taken and quenched in MeCN:water (1:1, by volume), followed by centrifugation and subsequent analysis using HPLC unless otherwise mentioned. Experiments were performed in triplicates. The initial rates were obtained from the linear range of product formation (or substrate consumption) over time, with the substrate(s) conversion limited to ≤ 20%. One unit (U) of enzyme activity represents the amount of enzyme required to release 1 µmol/min of product or consume substrate under the given assay conditions. Kinetic parameters ($V_{max}$, $K_m$) were determined by fitting the initial-rate data to Eq. (2) using non-linear least squares regression. The fitting was performed using Origin 2020 software (OriginLab, Northampton, MA, U.S.A.). In Eq. (2), V is the initial rate dependent on [S], $V_{max}$ is the maximum initial rate (µM/min), $K_m$ is the Michaelis constant (mM), and [S] is the initial substrate concentration (mM). The specific activity was calculated using the relationship $V_{spec} = V_{max}/[E]$, where [E] is the enzyme concentration in mg/mL. The rate constant $k_{cat}$ is determined from Eq. (3) where [E] is expressed as the molar concentration. The enzyme concentration ([E]) was determined based on the protein concentration, which was measured using absorbance at 280 nm and calculated using the protein-specific molar extinction coefficient of one active enzyme unit, i.e., the heterodimer for *Pu*CGE and enzyme subunit for *At*OGE, GlvA and BglT. [E] is calculated by multiplying [E] with the molar mass of the protein. Extinction coefficients and molar mass are given in the section "Photometric determination of protein and glycoside concentrations."

$$V = \frac{V_{max}[S]}{(K_M + [S])} \tag{2}$$

$$k_{cat} = \frac{V_{max}}{[E]} \tag{3}$$

***At*OGE and *Pu*CGE**. Initial rate measurements were carried out at 37 °C in potassium phosphate buffer (pH 6.0) supplemented with 5.0% DMSO. *Pu*CGE was preincubated with 5.0 mM MnCl$_2$ at 4 °C for 120 min prior to usage. For kinetic parameter determination, the following enzyme and substrate concentrations were used: $2.00 \times 10^{-4}$ mg/mL *At*OGE with 3-keto-phlorizin (0.1–6.0 mM), $2.50 \times 10^{-2}$ mg/mL *Pu*CGE with 3-ketopuerarin (0.1–3.6 mM), $5.00 \times 10^{-2}$ mg/mL *Pu*CGE with 3-keto-phlorizin (0.1–4.0 mM).

Initial rates for specific activity calculations were measured at 2.0 mM 3-keto-phlorizin, 3-keto-nothofagin and 3-keto-puerarin or at 16 mM 4NPα3ketoG and 4NPβ3ketoG. The enzyme concentrations were adjusted as required. Details are provided in Supplementary Figs. 37, 38, and 51.

To test *At*OGE activity towards 3-keto-*C*-glycosides (2.0 mM) and unoxidized phlorizin (2.0 mM), 0.50 mg/mL *At*OGE was used

(Supplementary Fig. 38). The activity of *At*OGE active site mutants (H134A, H189A) was measured with 3-keto-phlorizin (1.3 mM) and 2.40 mg/mL of enzyme. Additionally, reactions were conducted with *At*OGE variants being preincubated in 5.0 mM MnCl$_2$ prior to usage (see Supplementary Fig. 27).

The activity of *Pu*CGE variants was measured using 3-keto-phlorizin, 3-keto-puerarin and 3-keto-nothofagin (2.0 mM). *Pu*CGE$^\alpha$ and Y39F were used at a concentration of 0.50 mg/mL (Supplementary Fig. 57).

The importance of the Mn$^{2+}$ cofactor was evaluated by performing initial rate measurements using 2.0 mM substrate (3-keto-phlorizin for *At*OGE; 3-keto-phlorizin, 3-keto-nothofagin and 3-keto-puerarin for *Pu*CGE) under chelating conditions. To remove the cofactor from the enzyme, *At*OGE or *Pu*CGE were incubated with 1.0–10 mM EDTA for 120 min prior to reaction initiation. EDTA-treated *At*OGE was used at 0.01 mg/mL, and EDTA-treated *Pu*CGE was used at 0.05 mg/mL for 3-keto-puerarin and at 0.10 mg/mL for both 3-keto-phlorizin and -nothofagin (see Supplementary Fig. 40).

For measurement of the cleavage of 3-keto-disaccharides, a coupled assay with hexokinase was used instead of HPLC analysis. The assay solution consisted of 100 mM PIPES (pH 7.6), 4.7 mM ATP, 3.1 mM NAD$^+$, 4.9 mM MgCl$_2$, 1.5 U/mL hexokinase, 1.5 U/mL glucose 6-phosphate dehydrogenase. To 140 µL of assay solution, 2.0 mM of substrate was added and preincubated at 37 °C in a FLUOstar Omega-plate reader (BMG Labtech, Ortenberg, Germany) for at least 5 min. The reaction was initiated by addition of $3.00 \times 10^{-4}$ mg/mL *At*OGE or 1.20 mg/mL *Pu*CGE. It was ensured that *Pu*CGE or *At*OGE were added at rate-limiting concentrations (<1 U/mL) in the assay. Absorbance was followed continuously at 340 nm. The NADH extinction coefficient (340 nm, 6.22 AU mM$^{-1}$ cm$^{-1}$) was used to calculate the glucose release rate. Initial rates for kinetic parameter determination were measured using $3.00 \times 10^{-4}$ mg/mL *At*OGE with 3'-keto-cellobiose in the range of 0.2–6.0 mM. For activity tests 2.0 mM of 3'-keto-disaccharide were incubated with $3.00 \times 10^{-4}$ mg/mL *At*OGE or 1.20 mg/mL *Pu*CGE.

**GlvA and BglT**. Standard activity measurements used 1.0 mM phosphorylated substrate (phlorizin6P, nothofagin6P) in 50 mM HEPES buffer (pH 7.5; for nothofagin6P pH 6.5) supplemented with 0.5 mM MnSO$_4$, 0.3 mM NAD$^+$. The enzyme concentration used was 2.00 mg/mL.

For kinetic parameter determination at 37 °C, the phlorizin6P concentration was varied in the range from 0.1–4.0 mM and the enzyme concentration was reduced to 1.00 mg/mL. Activities (U) and other rates refer to aglycone released.

**GlycDH**. Standard activity measurements used 2.0 mM phlorizin in 100 mM potassium phosphate buffer (pH 5.7) supplemented with 8.0 mM of K$_3$[Fe(CN)$_6$], 10% DMSO and in the case of puerarin with 2.0 mM TCEP. Activities (U) refer to 3-keto-product formed. Enzyme concentrations of 0.02, 0.25 and 0.10 mg/mL were used for phlorizin, puerarin and nothofagin, respectively.

**BglK**. Standard activity measurements used 5.0 mM substrate in 50 mM HEPES buffer (pH 6.5) supplemented with 1.0 mM MgCl$_2$, 7.5 mM ATP and 10% (v/v) DMSO. The enzyme concentration used was 1.00 mg/mL BglK.

### Comparison of *At*OGE and *At*HYD as keto-glycoside deglycosidases

The substrate 3-keto-phlorizin (10 mM) was incubated in 10 mM potassium phosphate buffer (pH 7.0) with either *At*OGE or *At*HYD (both 0.30 mg/mL) in a total volume of 50 µL. Both enzymes were preincubated with MnCl$_2$ (5.0 mM, 120 min at 4 °C). No enzyme was administered for the negative control. The reaction was incubated at

37 °C and 650 rpm for 60 min. Typically, 2.0 μL samples were analyzed on TLC.

## Single-step deglycosylation reactions catalyzed by *Pu*CGE
In a one-pot approach, 1.0 mM unoxidized *C*-glycoside substrates (puerarin and nothofagin) were incubated in 10 mM potassium phosphate buffer (pH 6.0), supplemented with 8.0 mM $K_3[Fe(CN)_6]$. The reaction was initiated by addition of GlycDH and *Pu*CGE, each at 0.10 mg/mL. The reaction was performed at 2.0 mL scale at 37 °C and 650 rpm agitation. Samples taken (20 μL) were analyzed by HPLC.

## Conversion of 3-keto-phlorizin to glucose
In a one pot approach, 2.0 mM lyophilized 3-keto-phlorizin were dissolved in potassium phosphate buffer (10 mM; pH 7.0) supplemented with 4.0 mM NADH. The reaction was initiated by addition of Gfo oxidoreductase, *At*OGE and *At*HYD (each at 0.30 mg/mL). The reaction was incubated at 37 °C and 650 rpm (Eppendorf Thermomixer). The reaction was followed by TLC.

## Thin-layer chromatography (TLC)
About 2.0−10 μL of sample were applied onto a TLC plate (silica gel 60 F254 by Merck KGaA) and were separated with either a 9:1 mixture of MeCN and $H_2O$, or a 2:1:1 mixture of 2-BuOH, AcOH and $H_2O$. Subsequently, the dried plate was sprayed with either a detection solution consisting of 2.4% (w/v) *p*-anisaldehyde, 1.0% AcOH (v/v) and 3.4% (v/v) $H_2SO_4$ in 95% EtOH or 0.5% (w/v) thymol and 5.0% (v/v) $H_2SO_4$ in 95% EtOH. The sugars and flavonoids were visualized with heat treatment. Daidzein was detected with a UV-lamp at a wavelength of 256 nm.

## High-performance liquid chromatography (HPLC)
Samples (20 μL) were quenched in the same amount of MeCN, incubated on ice and centrifuged for 10 min at maximum speed in an Eppendorf 5415 R micro centrifuge to remove precipitated enzyme. Typically, 10 μL of sample were loaded on a Kinetex® Reversed Phase C18 column (Phenomenex, Aschaffenburg, Germany; 100 Å, 5.0 μm, 150 × 4.6 mm). Analytes were eluted in 20 mM potassium phosphate buffer (pH 5.9) containing 40 mM tetrabutylammonium bromide with a linear gradient of 25−70% MeCN. The gradient was run with a flow rate of 1.0 mL/min at 35 °C for 7.0 min, before residual compounds were removed for 1.0 min with 80% MeCN, upon 1.0 min of column regeneration. Compounds were detected at their respective absorption maxima of 256 nm (puerarin), 288 nm (nothofagin and phlorizin) and 310 nm (4NPαG, 4NPβG and their C3 oxidized or C6 phosphorylated forms). If a more detailed separation was required due to compounds with similar retention times, analytes were eluted in a 20 min long run with HPLC-gradient $H_2O$ containing 0.1% $H_2CO_2$ and a 10−50% linear gradient of MeCN was used. For the separation of puerarin-based compounds 0.05% of TFA instead of $H_2CO_2$ and a linear gradient of 0.0−50% was used. Subsequently, the column was regenerated for 4 min. The HPLC system employed in the study was either the Prominence 20 A system manufactured by Shimadzu (Kyoto, Japan) or the Agilent 1200 series system developed by Agilent Technologies (Santa Clara, CA, U.S.A.). The LabSolutions™ software v.5.6 from Shimadzu was used to operate HPLC instruments.

## *At*OGE and *Pu*CGE reaction product identification by $^1$H NMR
In general, 15 mM of 3-keto-substrate (4NPα3ketoG, 4NPβ3ketoG, 3′-keto-cellobiose, 3′-keto-maltose) was dissolved in potassium phosphate buffer prepared in $D_2O$ (pD 6.9). The total reaction volume was 700 μL. The reaction was initiated by enzyme addition; 0.04 mg/mL *At*OGE was used for 4NPα3ketoG and 4NPβ3ketoG reactions while 0.16 mg/mL *At*OGE was used for 3′-keto-cellobiose and 3′-keto-maltose. The reactions were incubated at 37 °C without shaking. $^1$H-NMR were measured before enzyme addition and at selected time points within a total incubation time of 20 h. $^1$H-NMR-measurements were performed as described in the section $^1$H-NMR analysis of purified keto-products.

In the case of substrates with poor solubility in water (3-keto-phlorizin, 3-keto-puerarin), the reactions were conducted at a scale of 10 to 30 mL. They comprised 2.0 mM of substrate dissolved in a 10 mM potassium phosphate buffer (pH 6.0) and 0.10 mg/mL of enzyme. The conversion was monitored by HPLC. At 90% conversion, the enzymes were removed by centrifugation 5975 × *g* with a Sartorius Vivaspin™ Turbo 15 concentrator tube used in an Eppendorf Centrifuge 5810R. The enzyme-free reaction mixture was concentrated under reduced pressure (20 mbar, 40 °C, 100 rpm) to a total volume of 0.5−2.0 mL in a Laborota 4000 efficient rotary evaporator (Heidolph Instruments). The sample was shock frozen and lyophilized for 16 h with a Christ Alpha 1−4 freeze dryer (bbi-biotech GmbH) using a Vacuubrand vacuum pump unit RZ 6. The lyophilisate was resuspended in 10% (v/v) DMSO-d6 and $D_2O$ to a total volume of 700 μL. The sample was analyzed using $^1$H-NMR and the same parameters as described in $^1$H-NMR analysis of purified keto-glycosides.

## $^1$H-NMR analysis of purified keto-glycosides
Lyophilized products were dissolved to 15 mM in 70 μL of DMSO-d6 (Armar AG, Döttingen, Switzerland) and 630 μL of $D_2O$ (Euroiso-Top GmbH, Saarbrücken, Germany). All samples were transferred to a 5.0 mm high-precision NMR sample tube (Promochem, Wesel, Germany) and measured in a Varian INOVA 500 MHz spectrometer (Agilent Technologies, $^1$H: 499.98 MHz) or a Bruker AVANCE III 300 spectrometer (Bruker, Rheinstetten, Germany, $^1$H: 300.36 MHz) at 30 °C using VNMRJ 2.2D software. $^1$H NMR spectra were measured on a 5.0 mm indirect detection PFG-probe and with pre-saturation of the water signal by a shaped pulse. Following standard pre-saturation sequence was applied: relaxation delay 2.0 s; 90° proton pulse; acquisition time 2.048 s; spectral width 8 kHz; number of points 32 k. Typically, 8−64 Scans were accumulated depending on the concentration of the samples. MestreNova v.14.0 (Mestrelab Research, Santiago de Compostela, Spain) was used for data processing.

## In situ $^1$H-NMR analysis of *At*OGE catalyzed 3-keto-phlorizin deglycosylation
Lyophilized 3-keto-phlorizin was dissolved to 22 mM in 10% (v/v) DMSO-d6 and 50 mM potassium phosphate buffer (pD 7.4) prepared in $D_2O$. The reaction was performed in a total volume of 700 μL (pD = pH + 0.4) using a 5.0 mm high-precision NMR sample tube. *At*OGE was re-buffered by washing the enzyme with 15-fold excess of 50 mM potassium phosphate buffer (pD 7.4) using a Sartorius Vivaspin™ Turbo 15 concentrator tube and centrifugation (5975 × *g* and 4.0 °C with an Eppendorf Centrifuge 5810R). The reaction was started by adding *At*OGE to a concentration of 0.35 mg/mL and incubated at 30 °C without agitation in the magnetic field. Measurements were taken with presets described in the section $^1$H-NMR analysis of purified keto-products.

## Crystallization of *At*OGE
The enzyme was prepared for crystallization by removing the N-terminal Strep-tag II with TEV protease restriction in 50 mM Tris/HCl buffer (pH 8.0) containing 1.0 mM dithiothreitol. The reaction was performed at room temperature for 120 min and continued at 4.0 °C for 16 h at a 50:1 ratio *At*OGE to TEV protease. Uncut protein and the removed Strep-tag II were captured by a 5.0 mL StrepTrap™ HP column equilibrated in loading buffer described in the Protein expression and Strep-trap purification section. The flow through containing untagged *At*OGE was concentrated to 2.0 mL in a Sartorius Vivaspin™ Turbo 15 concentrator tube using centrifugation at 5975 × *g* and 4.0 °C. The concentrate was loaded onto a size exclusion column (16 × 1000 mm; XR 16/100 column; GE Healthcare) packed with Sephadex G10 (exclusion limit <700 Da), equilibrated in filter sterilized

loading buffer containing HEPES (10 mM; pH 7.0), 150 mM NaCl and 0.1 mM TCEP and mounted onto an ÄKTA prime FPLC system (GE Healthcare). Elution was performed with same buffer at a flow rate of 1.0 mL/min, monitored at 280 nm and fractions containing the enzyme complex were pooled.

Screening for crystallization conditions was performed with an Oryx8 robot (Douglas Instruments Limited, Hungerford, UK) using the commercially available screens Morpheus®, JCSG Plus™ and SG1™ from Molecular Dimensions Limited (Holland, MI, U.S.A.) by the sitting drop vapor-diffusion method in 96-well plates. A stock solution of AtOGE at 10.0 mg/mL in loading buffer and 0.04 mM $MnCl_2$ was used for all crystallization experiments. Drops of 1.0 μL were pipetted for screens with a 1:1 ratio of protein and screening solution. The crystallization plates were incubated at 16 °C.

Initial crystals were obtained in JCSG-C3 condition containing 0.2 M ammonium nitrate, 20% (w/v) PEG 3350 and diffracted to 2.0 Å resolution. Data collection was performed at −173 °C (DESY P11 beamline, PETRA III, Hamburg, Germany[65]) on a crystal belonging to the space group $P2_12_12_1$ with unit cell parameters a = 36.6 Å, b = 93.7 Å, c = 123.7 Å, respectively. Data were processed and scaled using the XDS v.20220110 program package[66]. Molecular replacement was performed with Phaser v.2.8[67] using sequence guided ab-initio calculated model from RoseTTAFold server[52] as a search template. Extensive modification of the predicted model and subsequent refinement revealed two AtOGE molecules in the asymmetric unit with additional electron density for two manganese ions. Superposition of the experimental structure and the predicted RosettaFold model yields an RMSD of 0.987 (135/225 residues). The resulting model was manually completed in Coot v.0.9.8.7.1[68] and refined using the PHENIX v.1.19.2-4158 software suite[69]. The stereochemistry and geometry were analyzed using Molprobity v.4.5[70]. Data collection and processing statistics are summarized in Table S2. The atomic coordinates and structure factors have been deposited in the Protein Data Bank as entry 8BVK.

## Structural analysis of PuCGE and AtOGE
A comparison of AtOGE and other protein structures available in PDB was performed by Protein structure comparison PDBeFold[71] (http://www.ebi.ac.uk/msd-srv/ssm). For visualization and determination of the RMSD value AtOGE, AgCGD2, EuCGD and PuCGE (PDB code: 7DNN, 7EXB, and 7EXZ, respectively) were superimposed in PyMOL v.4.6 (DeLano Scientific, http://pymol.org).

## Docking and molecular dynamics simulations
The all-atom models for the binding modes of 3-ketopuerarin and 3-ketophlorizin to PuCGE and 3-ketophlorizin to AtOGE were obtained by a combined protocol of molecular docking and classical molecular dynamics (MD) simulations in explicit aqueous solvent. We considered the homodimer of AtOGE and the heterodimer of PuCGE for our simulations. The initial guess for the three complexes were generated by molecular docking with the Lamarckian genetic algorithm implemented in AutoDock 13.0 implemented in Yasara v. 18.2.7. The AMBER-15FB[72] was used for dockings. The ligand was generated using the Grade2 web server[73]. A simulation cell of 15 Å × 10 Å × 10 Å was placed around the bound $Mn^{2+}$ and 50 docking runs with a fully flexible ligand were performed. Results were clustered by RMSD of 2 Å. Visual inspection of the docking solutions together with the evaluation of the scoring function were used to select the best pose as initial guess for the MD simulations.

For the initial structures of AtOGE and PuCGE, we used their crystallographic structures (PDB ids. 8BVK and 7EXZ, respectively). In the case of AtOGE, we modeled the two missing loops in the X-ray structure (residues Asp42-Val46 and Thr198-Gly211) using the RCD+ v.2019 server[74]. We also included the missing C-terminus in AtOGE (Leu236-Glu253) after homology modeling with the server Phyre2.0[75]. The C-terminus in AtOGE was placed into the system after structural

superimposition with the X-ray solved structure. The AMBER force field parameters for 3-ketopuerarin and 3-ketophlorizin were computed on the optimized structures of the substrates. Importantly, 3-ketopuerarin was modeled as deprotonated on the oxygen on C-7 of the oxochromene moiety. First, the lowest energy conformers for both 3-ketopuerarin and 3-ketophlorizin were obtained via the Conformer-Rotamer Ensemble Sampling Tool (CREST) v.2.11.1[76] with an implicit solvation model for water. Then, the lowest-energy geometries were optimized with Density Functional Theory (B3LYP/def2-SVP[77]) with an implicit solvation model for water. Afterwards, the electrostatic potential was computed for both ligands following the standard procedure in Amber20 and finally, the RESP point charges were derived with antechamber. We used the metal ion $Mn^{2+}$ force field parameters described by Bradbrook et al.[78] The AMBER force field ff19SB[79], the GLYCAM-06j6[80] parameters for carbohydrates, and the monovalent ion parameters from Joung & Cheatham[81] and the Li/Merz ion parameters[82] of divalent to tetravalent ions for TIP4P/EW water model were used to model the protein, the sugar moieties, and the counterions, respectively. The protonation state of the side chains of His, Lys, Arg, Asp and Glu was assessed at pH 6.0 via the server H++[83]. The complexes 3-keto-phlorizin:PuCGE, 3-keto-puerarin:PuCGE, 3-keto-phlorizin:AtOGE were embedded in a box of OPC water molecules[84] and the system was neutralized with the addition of $Na^+$ by random substitution of water molecules.

The solvated systems were minimized in three consecutive steps where all the hydrogens of the system, everything except the solute, and finally, the entire system, were allowed to relax. The minimized systems were then heated up from 100 to 300 K in 50 ps under a NVP ensemble and applying a Langevin thermostat (friction coefficient = 1 ps$^{-1}$). Importantly, a harmonic force of 200 kcal mol$^{-1}$ Å$^{-2}$ was applied to all heavy atoms of the solute (enzyme + substrate) during the heating process. Afterwards, six equilibration steps were used to remove gradually the applied harmonic constraint and to change to an NPT ensemble. During the equilibration process, the distances between the ligands and the metal center $Mn^{2+}$ were constrained to avoid artefacts. Afterwards, the systems (3-keto-puerarin:PuCGE, 3-keto-phlorizin:PuCGE, 3-keto-phlorizin:AtOGE) were further simulated in three independent unrestrained MD simulations (3 × 0.30 μs, total time per system: 0.90 μs). Periodic boundary conditions were used and longe-range electrostatic interactions were treated with the smooth Particle Mesh Ewald (PME) method, with a cut-off of 10 Å for direct interactions. A representative structure of each of the substrate:enzyme (ES) complexes was selected from the larger cluster of conformers out of three computed with the hierarchical agglomerative (bottom-up) algorithm available in cpptraj[85]. All the MD simulations were run with Amber20[86] using pmemd.cuda and the geometry optimization of the substrates performed with Gaussian16 C.01[87].

In order to evaluate the effect of the active site environment on the puckering of the glycan part of the ligands, we measured the Cremer–Pople puckering descriptors[88] Q, theta and phi on both trajectories of the ligands in solution and bound to the metal center at the active site using the program geo.py included into SHARC v.2.0 (https://sharc-md.org). Therefore, we also simulated the ligands 3-keto-puerarin and 3-keto-phlorizin in a box of OPC water molecules at 300 K for 300 ns. In all cases, the simulations were run at the same force field level of theory as described above.

## Cloning of pET-Duet-1 pucge$^α$
The PuCGE$^α$ expression cassette was constructed performing one-step site-directed deletion mutagenesis of the pET-Duet-1 pucge$^α$ pucge$^β$ plasmid, removing the nucleotide pucge$^β$ region. Template DNA was isolated from E. coli TOP10F' with the Wizard® Plus SV Minipreps DNA Purifcation System (Promega, Fitchburg, WI, U.S.A.) and amplified by PCR using Q5 High Fidelity DNA Polymerase (New England Biolabs GmbH) and primer pairs with 5′-homologous overhangs to each other

(Table S7). The amplicons were treated with 1.0 μL of DpnI (Thermo Fisher Scientific, Inc.) at 37 °C for 120 min. Purification was performed with the Wizard® SV Gel and PCR Clean-Up System (Promega, Corporation) and transformation into electro competent *E. coli* TOP10F'. DNA sequencing was done at LGC Genomics (Berlin, Germany).

## Data availability

All relevant data are reported in the manuscript and in the associated Supplementary Information. All data are available from the corresponding author upon request. There is no restriction on data availability. The crystallographic data that support the findings of this study are available from the Protein Data Bank (http://www.rcsb.org). The coordinates and the structure factor amplitudes for the apo structure of *At*OGE were deposited under accession code 8BVK. Other structures *Ag*CGD2, *Eu*CGD and *Pu*CGE used for comparison are deposited under 7DNN, 7EXB and 7EXZ, respectively. Sequence information on proteins used within this study is available on the NCBI database (https://www.ncbi.nlm.nih.gov/). GlycDH and TAT-pathway signal protein from *Rhizobium sp.* GIN611, *At*OGE, *At*HYD and Gfo oxidoreductase from *A. tumefaciens*, *Pu*CGEα and *Pu*CGEβ from PUE strain, BglK from *K. pneumoniae*, GlvA from *B. subtilis* and BglT from *T. maritima* are registered under accession numbers AEX01166.1, AEX01167.1, WP_003515232.1, WP_038490972.1, WP_038490970.1, BBG22494.1 and BBG22495.1, AAK58463.1, WP_088325462.1, and WP_004079953.1, respectively. Other Source data are provided with this paper. Molecular dynamics simulation trajectories are deposited on Zenodo (https://zenodo.org) under the creator https://doi.org/10.5281/zenodo.8319647. Source data are provided with this paper.

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

## Acknowledgements

We thank Saskia Hofer (Institute of Biotechnology and Biochemical Engineering, Graz University of Technology) for enzyme expression and isolation and Professor Hansjörg Weber (Institue of Organic Chemistry, Graz University of Technology) for NMR measurements. We acknowledge DESY (Hamburg, Germany), a member of the Helmholtz Association HGF, for the provision of experimental facilities (beamline P11). Beam time was allocated for proposal BAG-20200791 EC. Funding by the Austrian Science Fund (FWF): CATALOX (doc.funds46) as the main funding and doc.fund grant DOC-130 for TPK are gratefully acknowledged.

## Author contributions

J.B., design of study; experiments and data analysis; writing of paper. M.P., design of study; experiments and data analysis. A.J.E.B., mechanistic schemes and NMR discussion. K.K. and T.P.-K., crystallographic experiments and data analysis thereof. P.A.S.-M., molecular dynamics simulations and data analysis thereof; B.N., design of study; funding acquisition; discussion; writing of paper.

## Competing interests

The authors declare no competing interests.
