## [Peer Review File · Nature Communications]

REVIEWER COMMENTS

Reviewer #1 (Remarks to the Author):

This manuscript by Bitter et al. details a biochemical study into the cleavage of O- and C-glucosides that occurs via an elimination mechanism following oxidation of the C3-hydroxyl in beta-glucosides by a separate enzyme.

1) The authors write: "Among glycoside hydrolases, family GH4 enzymes are unusual in lacking strict substrate discrimination according to glycosidic bond alpha or beta stereochemistry", but the enzyme in this reference has a natural substrate of 6-phosphomaltose, and it can only hydrolyze beta 6-phosphoglucosides that have activated leaving groups. Whereas, the natural substrates for AtOGE contains an activated aryl beta-glucoside. So the comparison with a GH4 enzyme using a non-natural diastereomeric substrate with the enzyme AtOGE using its natural substrate is, in my opinion, not a useful comparison. A more useful comparison would have been to a beta-phosphoglucosidase, e.g., see Varrot, et al NAD⁺ and metal-ion dependent hydrolysis by family 4 glycosidases: Structural insight into specificity for phospho-b-D-glucosides. *J. Mol. Biol.* 2005, 346, 423.

2) The mechanism of GH4 enzymes (Figure S2) is shown with acid catalysis for the ejection of the aglycone, however, Sannikova et al. (Both chemical and non-chemical steps limit the catalytic efficiency of family 4 glycoside hydrolases. *Biochemistry* 2018, 3378) argue that no proton catalysis is needed for an E1CB reaction. This possibility should be included in the current MS as loss of an alkoxide from an enolate to give an alpha,beta-unsaturated ketone is the reaction that occurs in solution, also, for GHs aryloxide aglycone departure either does not involve proton assistance or it is only weakly catalyzed. Such an analysis explains the ability of GH4 enzymes to hydrolyze thioglycosides.

3) The authors state near the bottom of page 3: "Since the buffer conditions evidently control the 2- and 3-keto-product ratio at apparent equilibrium, ...", a statement that is incorrect. The equilibrium constant for this isomerization will be independent of pH/buffer/enzyme concentration. However, the rate at which equilibrium is reached will depend on the buffer and its concentration. I find it curious that for the oxidation reactions in HEPES (Fig 2E) that the authors used a much higher (10x) enzyme concentration than in Fig 2D, it is possible that the enzyme also catalyzes the isomerization, was this checked?

4) We are given a rate constant ($4.5E-6$ s⁻¹; line 169) but I can't find the experimental details in the manuscript. I presume that the rate constants refers to a pH of 6.5. Was this number calculated from a single data point at 16 hours (HPLC trace in figS16a)?

5) I was curious if the authors modeled the binding of 3-keto substrates with the manganese ion coordinated between O-3 and O-2, by removing the bound Mn-OH₂. Coordination of a divalent cation to the C2-OH group (as noted for GH4 enzymes) acidifies the C2-H for the elimination reaction. I'd be surprised if this interaction was not operative in AtOGE, some discussion on this point is required.

6) The enzymatic kinetics reported are single substrate concentrations, as a result there is no information on the kinetic constants k_{cat} and k_{cat}/K_m .

7) Figure 6C, the structures in brackets are not "tautomers" they have the same connectivity. Rather it is the equilibrium arrows labeled with PuCGE that is the tautomerization. The authors should consider whether the enzyme perturbs the equilibrium constant for tautomerization, which for phenol is about $10E-12$ (for example by distortion of the planar aromatic ring), as well as accelerating the phenol <- cyclohexadienone interconversion. Again, the authors should discuss whether the Mn²⁺ binds to the C2-OH as well as the C3=O.

In summary, at the present time, I'm not convinced that the current manuscript reports significant new results for publication in *Nature Communications*.

Typographical errors include:

Lines 571 and 574 "p-nitrophenol-alpha/beta-D-glucose" should be 4-nitrophenyl alpha- and beta-D-glucopyranosides", there is no hyphen between the aglycone and the glycoside, please correct throughout the document. The correct IUPAC name for (pNPalpha3ketoG) is 4-nitrophenyl alpha-D-arabino-hexopyranosid-3-ulose: Carbohydr. Res. 1997, 297, 1-92.

Reviewer #2 (Remarks to the Author):

In this work by Bitter and co-workers they describe the identification of one and characterisation of two enzymes that catalyse the degradation of 3-keto sugars, as part of a multi-enzyme bacterial pathway for degradation of flavanoid glycosides. Notable is the difference in activity of these enzymes on C-linked glycosides, as opposed to the more common O-linked, and the authors do an excellent job of drilling down into the differences between these enzymes that allows one to efficiently cleave these C-C bonds while the other cannot. They present extensive enzyme kinetics, supported by TLC, HPLC, and NMR monitoring and characterisation, as well as phylogenetic analysis that shows such gene clusters are likely common in soil and gut microbiota, and solve a crystal structure of their newly discovered enzyme. Overall this is an excellent piece of work that is well carried out and a significant advance to the field, improving our understanding of the workings of eliminative mechanisms in glycoside degradation. I recommend publication.

Some minor suggestions to improve the manuscript:

-page 5 lines 181-193, the significance of the buffer effect was not immediately apparent to me. This is a surprising result, that buffer would change the equilibrium so drastically with no change in pH. The text could better introduce this, for example in line 183 instead of "...strong buffer effect on the product formation..." to make this more precise, e.g.

"...strong buffer effect on the product regiochemical equilibrium..."

-I find figure S6 much more informative than figure 2, as it directly shows the above-mentioned equilibration. The authors could consider swapping these two figures.

-The choice of yellow for emphasis across figures is unfortunate, as it makes some difficult to read. Consider making at least the text black to improve this.

-The title on page 12 line 400 is slightly misleading – GH4 hydrolases show low activity for O-glycoside cleavage of these substrates (as opposed to the no activity for C-glycoside hydrolysis). The title could be read to mean these are poor O-glycoside hydrolases globally.

-Kinetic parameters in Figs S46d and S48d are poorly defined by the data (no plateau, and no datapoints in the linear region, respectively). These are not over-interpreted, so likely not worth additional data, but perhaps warrant a caveat in the caption.

Reviewer #3 (Remarks to the Author):

The article represents an elegant study on a new enzyme (β -eliminase AtOGE) with unrelated sequence identity and specific for O-glycosidic substrate. Quite robust work to prove the mechanistic function of this novel enzyme. The methodology involves a range of analytical chemical, biochemical, biophysical, molecular, and structural biology approaches that contribute to a large volume of results well-presented throughout the main text and supplementary material. I suggest some modifications and explanations before publication in this remarkable journal.

- Introduction is too long and needs to be more focused

- Results present many aspects of methodology

- The role of the ion is not fully clear (despite being reported conserved in the folding with others proteins); a mutagenesis study would be interesting in order to understand the role of ions in the function. The authors comment that likely Mn^{2+} lies at the active site of the X-tal structure. Does the enzyme show activity in other types of divalent ions? There is a lack of a functional study of the enzyme in solution with different ions, EDTA, including performing a chelex step on the purified protein.

- The authors comment that the enzyme is a homodimer, but I wonder if the work presents evidence based on methods to study proteins in solution (no figure was indicated in the text – line 247). Has this conclusion been made based solely on the structural data? A better understanding of the existence of the dimer in solution with adequate methodologies and its role in the enzyme function is required.
- In the genomic context of *Agrobacterium tumefaciens*, there was a lack of insight into how this new enzyme relates to others of the same operon. Have other enzymes of this operon already been characterized? For the complete understanding of the metabolism of natural products, this is fundamental, and even the deletion of this gene in native bacteria.

Response to reviewers/Revision

18-May-2023

Please find below detailed response to the points raised in the review of our manuscript. We are grateful for the consideration of our manuscript and the constructive criticism provided. Our responses are formatted in blue color and are additionally marked with Response. We describe our position with respect to each point raised and explain the changes made upon revision. Overall, we believe to have addressed all points in an exhaustive manner.

Reviewer Comments

Response in general

We are grateful for a thorough and constructive review done by the three Reviewers. More detailed responses are provided below. We agree in general with the points of criticism and made significant effort to provide answers in the form of new and more detailed evidence.

Main changes at a glance: We included results for the β -specific phosphoglucosidase BglT for better comparability with AtOGE and PuCGE (**p14**); kinetic parameters determined from full-fledged Michaelis-Menten analysis at different substrate concentration ($n=3$) for PuCGE with 3-keto-*O*- and -*C*-glycosides, for AtOGE with 3-keto-*O*-aryl-glucoside and 3-keto-disaccharide, and for GH4 enzyme with phospho-*O*-glycoside (**Table S4**); mutation study of AtOGE in the Mn^{2+} -coordinating sphere (**Supplementary Figure 27**); full study by molecular dynamics simulation for the refinement of the docking poses, to allow for a more detailed discussion of the reaction mechanism (**p9-10**); AtOGE reaction with α - or β -linked disaccharides for insight in the stereochemical course of the glycoside cleavage (**p12**). To provide the fresh experimental evidence necessitated that several new methods be used. The Methods section is therefore expanded considerably (e.g., initial rate analysis; C3-specific disaccharide oxidation, isolation of 3-keto-disaccharide and analysis of 3-keto-disaccharide deglycosylation; molecular dynamics simulations). The Supplementary Information was expanded by including the new results in full detail.

Reviewer #1 (Remarks to the Author):

This manuscript by Bitter et al. details a biochemical study into the cleavage of *O*- and *C*-glucosides that occurs via an elimination mechanism following oxidation of the C3-hydroxyl in beta-glucosides by a separate enzyme.

1) The authors write: "Among glycoside hydrolases, family GH4 enzymes are unusual in lacking strict substrate discrimination according to glycosidic bond alpha or beta stereochemistry", but the enzyme in this reference has a natural substrate of 6-phospho-maltose, and it can only hydrolyze beta 6-phosphoglucosides that have activated leaving groups. Whereas, the natural substrates for AtOGE contains an activated aryl beta-glucoside. So, the comparison with a GH4 enzyme using a non-natural diastereomeric substrate with the enzyme AtOGE using its natural substrate is, in my opinion, not a useful comparison. A more useful comparison would have been to a beta-phosphoglucosidase, e.g., see Varrot, et al NAD⁺ and metal-ion dependent hydrolysis by family 4 glycosidases: Structural insight into specificity for phospho- β -D-glucosides. J. Mol. Biol. 2005, 346, 423.

Response: We thank the Reviewer for the comment. We agree with the Reviewer that the beta-phosphoglucosidase BglT is a relevant candidate for comparison with AtOGE. We do not agree with the Reviewer that the comparison with the alpha-phosphoglucosidase GlvA was not a useful comparison. Irrespective of expert opinion regarding this point, the best way to deal with the matter is to show both enzymes. This is what we do in the revised version and the results clarify the point. Both enzymes are inactive with 6-phospho- β -C-glucoside (6-phospho-nothofagin) and show low activity with the relevant 6-phospho- β -O-glucoside (6-phospho-phlorizin). There is no substantial difference in activity for the beta-phosphoglucosidase BglT and alpha-phosphoglucosidase GlvA. We also noticed that in the original manuscript, our discussion was perhaps too strongly focused on GH4 enzyme reactivity (efficiency) for 6-phospho- β -O-glucoside cleavage. The main point is that activity with the 6-phospho- β -C-glucoside is lacking, now shown for both types of GH4 enzyme. In summary, we believe that critique about our use of GH4 enzymes for comparison with AtOGE has been eliminated fully, and we consider our response to this point to be final.

Change: Activity of BglT with phosphorylated O- and C-glucosides reported (**Supplementary Figure 61**); kinetic parameters of BglT with phlorizin-6-P determined (**Supplementary Table 4**). **Supplementary Table 6** was complemented with the BglT sequence. The main text section on reactivity of GH4 enzymes includes the new data. Discussion was revised to focus on lack of activity with the 6-phospho- β -C-glucoside substrate. Since the same experimental protocols are used for reactions of GlvA and BglT, changes in the Methods section are only minor.

2) The mechanism of GH4 enzymes (Figure S2) is shown with acid catalysis for the ejection of the aglycone, however, Sannikova et al. (Both chemical and non-chemical steps limit the catalytic efficiency of family 4 glycoside hydrolases. *Biochemistry* 2018, 3378) argue that no proton catalysis is needed for an E1CB reaction. This possibility should be included in the current MS as loss of an alkoxide from an enolate to give an alpha,beta-unsaturated ketone is the reaction that occurs in solution, also, for GHs aryloxide aglycone departure either does not involve proton assistance or it is only weakly catalyzed. Such an analysis explains the ability of GH4 enzymes to hydrolyze thioglycosides.

Response: We agree with the Reviewer and notice in the revision that our discussion of the GH4 enzymes was in need of revision. Here, we realized a dilemma regarding the depth of discussion needed in our manuscript. The mechanism of the GH4 enzymes has been studied in great detail by highly advanced methods. In our interpretation of the existing evidence, there may be a continuum of mechanisms regarding the involvement of catalytic assistance in the step of expulsion of the leaving group. As shown clearly by the Bennet group, there are situations in which the leaving group departure may not involve facilitation by a protonic group. Our own results showing limited stability of 3-keto- β -O-glucosides to nonenzymatic decomposition support the idea of spontaneous departure of leaving group. However, our direct comparison of 3-keto-substrate decomposition in the absence and presence of AtOGE also shows acceleration of the substrate cleavage by a large factor of 10^7 -fold. Our conclusion at this stage is that the catalytic factors of the AtOGE reaction warrant more detailed mechanistic investigation for clarification in a later study. To be clear, this must in no way be interpreted as a limitation of the current research which is about the discovery of a new type of β -eliminase and the relationship of this enzyme with related β -eliminases for the cleavage

of C-glycosides. We suggest that one should not lose sight of the overall achievements of the study and return to this point when responding to the concluding remark of Reviewer #1.

Returning to the dilemma mentioned above, we reduced the level of depth of discussion of GH4 enzymes in the Introduction. [This also helps to comply with the criticism of Reviewer #3 that the Introduction is too long.] We provide a description of the GH4 cleavage mechanism without general acid catalytic assistance to leaving group expulsion in **Supplementary Figure 2b**. In the Discussion, we briefly discuss the point that the leaving group departure might occur with or without catalytic facilitation. Based on the combined evidence of enzyme structure and reactivity analyses with α - and β -configured 3-keto-*O*-glucoside substrates (note: the 4-nitrophenol substrate series was complemented by 3-keto-derivatives of the disaccharides cellobiose and maltose; only the β -configured cellobiose is reactive), we suggest a *tentative* mechanism that involves catalytic facilitation by the histidine residue that is also involved in deprotonation at C2 during enolate formation. The writing is done with all due caution, but the authors reserve the right to deliver a mechanistic proposal that is considered to be the most plausible in light of the evidence presented. As any mechanistic proposal (see the role of acid catalysis in glycosidic bond cleavage of the GH4 mechanism as an example; Sannikova et al. *Biochemistry* 2018, 57, 3378 as compared to Chakladar et al. *Biochemistry* 2010, 50, 4298), the current one may be subject to revision given the appearance of new evidence, but as the research stands, Figure 7B and discussion are consistent and plausible.

Change: Molecular dynamics simulations provide new and better supported evidence of substrate binding in AtOGE (**Figure 5a**, **Supplementary Figures 31** and **32**; sugar ring pucker analysis in **Supplementary Figure 30**). The stereochemical preference of AtOGE for reaction with α - and β -configured substrates is obtained from study of substrate pairs featuring activated (4-nitrophenol) and non-activated leaving groups (glucose). See **Supplementary Figures 47-50** and **Supplementary Figure 51-53** for new results. The Discussion was revised to show the full diversity of mechanisms considered for GH4 enzymes and gives an updated description of the proposed AtOGE mechanism. The Methods were changed as required.

3) The authors state near the bottom of page 3: "Since the buffer conditions evidently control the 2- and 3-keto-product ratio at apparent equilibrium, ...", a statement that is incorrect. The equilibrium constant for this isomerization will be independent of pH/buffer/enzyme concentration. However, the rate at which equilibrium is reached will depend on the buffer and its concentration. I find it curious that for the oxidation reactions in HEPES (Fig 2E) that the authors used a much higher (10x) enzyme concentration than in Fig 2D, it is possible that the enzyme also catalyzes the isomerization, was this checked?

Response: We agree with the Reviewer. The relevant part of the manuscript required revision. The results were not clear due to several parameters varied at a time in different experiments, and the writing was not clear as well. Additional difficulty was that the oxidized nothofagin and puerarin do not show identical behavior regarding keto group 2,3-isomerization. We performed additional experiments to obtain a consistent set of data. The updated evidence supports the conclusion that in the case of the keto-puerarin the equilibrium between the 2- and 3-keto-forms is reached quickly in solution. At all times during the reaction, the ratio between the two forms is the same within limits of error. In the case of the keto-nothofagin, the observed composition of 2- and 3-keto-forms during the reaction

is strongly dependent on the pH. At a low pH of 5.0, the 3-keto-form is preferred whereas at higher pH the 2-keto-form becomes dominant. Experiments done with isolated keto-phlorizin product comprised of 3-keto-form in excess show the spontaneous 3→2 isomerization in the absence of enzyme. Experiment done at high enzyme concentration reveals fast formation of 3-keto-product until almost all substrate is consumed and the 3→2 isomerization happens at a rate comparable to that in the experiment without enzyme. Overall, therefore, the plausible proposal is that GlycDH oxidizes the C-glycosides specifically at C3 and the mixture of 2- and 3-keto-product arises from spontaneous isomerization at a rate dependent on compound and conditions used (e.g., pH).

Change: The new data were added and the relevant section was rephrased (p5). **Supplementary Figure 9** provides the results in full detail. The Methods are revised.

4) We are given a rate constant ($4.5E-6 \text{ s}^{-1}$; line 169) but I can't find the experimental details in the manuscript. I presume that the rate constants refers to a pH of 6.5. Was this number calculated from a single data point at 16 hours (HPLC trace in figS16a)?

Response: We agree with the Reviewer that the rate constant for the spontaneous reaction was not well documented. It was from a single experiment and therefore not well supported by evidence. We therefore performed initial rate studies of spontaneous cleavage of 3-keto-phlorizin at 3 different concentrations. From these studies, an apparent first-order rate constant of nonenzymatic cleavage of substrate is obtained. The rate constant now determined ($k_{\text{non}} = 8.6 \times 10^{-6} \text{ s}^{-1}$) is similar to the one earlier estimated (30% difference). The k_{non} is now used for comparison with the enzymatic k_{cat} . The comparison is relevant and stands as is, even though we are aware that a detailed exploration of the catalytic rate acceleration by AtOGE may require further studies. However, our main point, that the enzyme speeds up the cleavage of the substrate to an extent that could easily be relevant biologically, is supported very well and unambiguously by the results shown.

Change: The described rate constant is now calculated from a larger set of adequate initial rate data (**Supplementary Figure 54**) and experimental details are introduced in the Methods section (“Measuring spontaneous deglycosylation of 3-keto-phlorizin”).

5) I was curious if the authors modeled the binding of 3-keto substrates with the manganese ion coordinated between O-3 and O-2, by removing the bound Mn-OH₂. Coordination of a divalent cation to the C2-OH group (as noted for GH4 enzymes) acidifies the C2-H for the elimination reaction. I'd be surprised if this interaction was not operative in AtOGE, some discussion on this point is required.

Response: In the AtOGE and PuCGE crystal structures, the Mn^{2+} is coordinated by 5 ligands, four coordinating protein residues and a water molecule. The original manuscript contained the error that the Mn^{2+} was drawn to have 6-fold coordination sphere. The error was corrected. Now, the point of the Reviewer is very interesting and valid. The GH4 enzymes (BglT, GlvA) for which structure of the product complex is available (i.e., enzyme bound with NAD^+ , Glc6P and Mn^{2+}), the metal also shows 5-fold coordination. In an apo-structure of GH4 enzyme (GglA with Mn^{2+}), the metal shows less coordination from protein and ligand groups but there seem to be water molecules that substitute in these interactions. In AtOGE, only

one water is present and the MD simulation results suggest that this will be replaced by the 3-keto-group of the substrate.

Figure: Mn^{2+} coordination in GH4 enzymes and in AtOGE. a BglI in complex with NAD^+ , Glc6P and Mn^{2+} (1UP6A). b GlvA in complex with NAD^+ , Glc6P and Mn^{2+} (1U8X). c AgIA in complex with Mn^{2+} (3U95). d AtOGE in complex with Mn^{2+} .

Our docking experiments combined with extensive MD simulations removed the water molecules from the AtOGE active site, including the one coordinated to the Mn^{2+} . This is a standard procedure in the computational analysis. Our MD simulations gave the consistent result that in both enzymes AtOGE and PuCGE, the 3-keto-substrate was always bound to the Mn^{2+} via the 3-keto group and the 2-OH was engaged in interactions with protein residue(s). No conformation was sampled (not even in small proportion) that had the 2-OH coordinating to the metal. We agree with the Reviewer that substrate binding under coordination of the 2-OH to the Mn^{2+} would be one way of the enzyme to enhance the reactivity at the C2 for deprotonation. However, the use of hydrogen bonding is an alternative way for the enzyme, probably equally efficient as metal coordination, to achieve the same. In summary, therefore, we propose the enzyme complex conformers obtained from the MD simulations (**Figure 5**) as plausibly functional/reactive. The evidence shown does not completely exclude the possibility that the 2-OH might also be coordinating. However, any coordination of the 2-OH would require a protein ligand to be displaced.

Change: Extensive MD simulations were performed with both enzymes. For PuCGE, both *O*- and *C*-glycoside substrate was modeled. The results are described on **p9** and **10**, and are graphically depicted in **Figure 5** and **Supplementary Figures 29 - 36**. The respective MD simulations protocols were added to the Methods section on **p26 - 27**.

6) The enzymatic kinetics reported are single substrate concentrations, as a result there is no information on the kinetic constants k_{cat} and k_{cat}/K_m .

Response/Change: We agree with the Reviewer. We determined kinetic parameters ($n=3$) for PuCGE with 3-keto-phlorizin and 3-keto-puerarin; for AtOGE with 3-keto-phlorizin and 3-keto-cellulose; and for BglT with phlorizin-6-P. Representative substrate ranges and enzyme concentrations were used to ensure initial rates determined from the linear range of product formation with time. The obtained values were added to **Supplementary Table 4** and kinetic values and specific activities were updated in the main text.

7) Figure 6C, the structures in brackets are not "tautomers" they have the same connectivity. Rather it is the equilibrium arrows labeled with PuCGE that is the tautomerization. The authors should consider whether the enzyme perturbs the equilibrium constant for tautomerization, which for phenol is about $10E-12$ (for example by distortion of the planar aromatic ring), as well as accelerating the phenol \rightleftharpoons cyclohexadienone interconversion. Again, the authors should discuss whether the Mn^{2+} binds to the C2-OH as well as the C3=O.

Response: We thank the Reviewer for pointing out the error regarding the indication of tautomerization. At this stage, we cannot speculate about assistance from the enzyme to change the tautomer equilibrium. For sure, the placement of the leaving group in a protein environment that accommodates the neighboring phenolic OH properly (see the Discussion of an oxyanion binding pocket) will be important. The extent of shift in equilibrium is not possible to estimate based on the evidence available. The point about the C2-OH involved in coordination was addressed above.

Change: The mechanism was updated and **Figure 7** was corrected.

In summary, at the present time, I'm not convinced that the current manuscript reports significant new results for publication in Nature Communications.

Response: We interpret the comment as an invitation to improve our manuscript. As a general comment on the study, however, we do not agree. Our impression is that the Reviewer might arrive at this (in our reading) negative conclusion based on giving excessive weight on her/his assessment of mechanistic details. The current study has discovered a new enzyme for 3-keto-*O*-glycoside cleavage from a previously unknown pathway broadly distributed in plant-associated bacteria. Elucidation of structural and functional characteristics of the OGE defines relationship to enzyme able to cleave *C*-glycosidic bonds. The comparison between OGE and CGE shows the CGE also for the cleavage of *O*-glycosides. This has enabled the authors to use mutagenesis to identify interactions specifically required for the *C*-glycoside cleavage. The evidence supports an updated mechanistic proposal for the *C*-glycoside cleavage. We certainly agree on the Reviewer's points about being clearer and

more precise on the mechanistic details of GH4 enzymes. We also agree on the need to clarify with more/better evidence and clearer and more cautious writing, especially regarding mechanistic claims. As summarized above, a substantial effort has been made in the revision. We retain the claim that overall, this study is highly significant in the progress made.

Typographical errors include: Lines 571 and 574 "p-nitrophenol-alpha/beta-D-glucose" should be 4-nitrophenyl alpha- and beta-D-glucopyranosides", there is no hyphen between the aglycone and the glycoside, please correct throughout the document. The correct IUPAC name for (pNPalpha3ketoG) is 4-nitrophenyl alpha-D-arabino-hexopyranosid-3-ulose: Carbohydr. Res. 1997, 297, 1-92.

Response/Change: We thank the Reviewer for this indication and changed all nomenclature including falsely labeled "p-nitrophenol" compounds accordingly to the correct IUPAC name.

Reviewer #2 (Remarks to the Author):

In this work by Bitter and co-workers they describe the identification of one and characterisation of two enzymes that catalyse the degradation of 3-keto sugars, as part of a multi-enzyme bacterial pathway for degradation of flavanoid glycosides. Notable is the difference in activity of these enzymes on C-linked glycosides, as opposed to the more common O-linked, and the authors do an excellent job of drilling down into the differences between these enzymes that allows one to efficiently cleave these C-C bonds while the other cannot. They present extensive enzyme kinetics, supported by TLC, HPLC, and NMR monitoring and characterisation, as well as phylogenetic analysis that shows such gene clusters are likely common in soil and gut microbiota, and solve a crystal structure of their newly discovered enzyme. Overall this is an excellent piece of work that is well carried out and a significant advance to the field, improving our understanding of the workings of eliminative mechanisms in glycoside degradation. I recommend publication. Some minor suggestions to improve the manuscript:

Response: We thank the reviewer for the overall positive assessment of our study.

-page 5 lines 181-193, the significance of the buffer effect was not immediately apparent to me. This is a surprising result, that buffer would change the equilibrium so drastically with no change in pH. The text could better introduce this, for example in line 183 instead of "...strong buffer effect on the product formation..." to make this more precise, e.g. "...strong buffer effect on the product regiochemical equilibrium..."

Response/Change: We agree with the Reviewer on the need to have this section revised. The point here is very similar to the point #3 of Reviewer 1. We have performed the new experiments as already discussed above. We think that the evidence shown in the revised manuscript is convincing. Please also revisit the response to Reviewer 1 under #3 and changes made accordingly.

-I find figure S6 much more informative than figure 2, as it directly shows the above-mentioned equilibration. The authors could consider swapping these two figures.

Response/Change: We thank the reviewer for this input and accordingly assembled **Figure 2** of time courses for *O*- and *C*-glycoside oxidation instead of HPLC traces. The HPLC traces can be now found in **Supplementary Figure 6** and a separate **Supplementary Figure 9** addresses explicitly the 2- and 3-keto isomerization in nothofagin.

-The choice of yellow for emphasis across figures is unfortunate, as it makes some difficult to read. Consider making at least the text black to improve this.

Response/Change: We changed the yellow color to darker orange throughout the document for better readability and emphasis.

-The title on page 12 line 400 is slightly misleading – GH4 hydrolases show low activity for *O*-glycoside cleavage of these substrates (as opposed to the no activity for *C*-glycoside hydrolysis). The title could be read to mean these are poor *O*-glycoside hydrolases globally.

Response/Change: We agree with the Reviewer. It may also have been a point of Reviewer #1, and we realized that our writing might be easily misunderstood as implying that GH4 enzymes are generally poor catalysts of *O*-glycoside cleavage. We do not imply anything in that regard. We changed the title of the subsection to “Family GH4 glycoside hydrolase do not cleave phloretin *C*-glycoside”, so it does not imply bad activity with *O*-glycosides in general.

-Kinetic parameters in Figs S46d and S48d are poorly defined by the data (no plateau, and no datapoints in the linear region, respectively). These are not over-interpreted, so likely not worth additional data, but perhaps warrant a caveat in the caption.

Response/Change: We agree that the kinetic parameters for the two enzymes could have been defined better, with more data taken in the relevant regions. However, in light of the whole, the importance of these parameters is really minor. We therefore follow the suggestion and indicate in the legend of the **Supplementary Figure 60** that the parameters should be considered as estimates. The kinetic parameters for the BglK (kinase used to prepare the 6-phospho-glucosides) are not relevant and have been removed.

Reviewer #3 (Remarks to the Author):

The article represents an elegant study on a new enzyme (β -eliminase AtOGE) with unrelated sequence identity and specific for *O*-glycosidic substrate. Quite robust work to prove the mechanistic function of this novel enzyme. The methodology involves a range of analytical chemical, biochemical, biophysical, molecular, and structural biology approaches that contribute to a large volume of results well-presented throughout the main text and supplementary material. I suggest some modifications and explanations before publication in this remarkable journal.

Response: We thank the reviewer for this substantial complement on our research.

- Introduction is too long and needs to be more focused

Response: The Introduction was shortened by about 20% in word count (847 compared to 1047). The shortening was achieved in four ways. (1) The writing was made more concise. (2) The information on GH4 enzymes was decreased in the level of detail. (3) The preview on the results was shortened. And (4) the general introduction on polyphenols/flavonoids was shortened. Generally, however, we believe that none of the lines of introduction is superfluous or unnecessary. So, we keep the general structure of the Introduction.

Change: Shortening of the Introduction as explained.

- Results present many aspects of methodology

Response: We tried to identify point where the use of details from experiment/method may have been excessive. To be frank, we didn't find many places where this would have seemed necessary. As mentioned in the response to Reviewer 1 (#3) and Reviewer 2 (first point), one part in need of improvement had to do with the formation of keto-product and 2,3-isomerization thereof. The experiments done were not entirely systematic and this has resulted in the requirement to report numerous details of the individual experiment. This part has been revised carefully. In other places of the manuscript, we have revised the wording and left away reaction conditions when this was clear from the Methods.

Change: Results part was checked and small changes were made throughout. The part revised most substantially was that reporting the substrate oxidation by GlycDH.

- The role of the ion is not fully clear (despite being reported conserved in the folding with others proteins); a mutagenesis study would be interesting in order to understand the role of ions in the function. The authors comment that likely Mn^{2+} lies at the active site of the X-tal structure. Does the enzyme show activity in other types of divalent ions? There is a lack of a functional study of the enzyme in solution with different ions, EDTA, including performing a chelex step on the purified protein.

Response: We agree with the Reviewer on the interest in the role of the coordinating residues. Given the need to prioritize in our study which starts to reach the limit of what can be accommodated in a single paper, we decided on the following course of revision. A single residue of the Mn^{2+} coordination sphere of AtOGE (His189) was replaced by a non-coordinating residue (Ala). The resulting variant was expressed, purified and analyzed. It was shown to retain less than 0.1% of wildtype activity. Now, one could certainly expand the characterization of the H189A variant and include further variant at position His189 and other coordinating positions. However, the metal dependence of AtOGE activity is already well supported by experiments using EDTA as metal chelator (**p11** and **Supplementary Figure 40**). Moreover, the literature shows variants PuCGE with substitution of coordinating residues by residues unable to fulfil the same function. Variants of PuCGE, featuring a disruption of their Mn^{2+} coordination sphere, are not active at all or they show very low activity. We therefore felt that further efforts to demonstrate the functional importance of Mn^{2+} site (which is already well supported by the combined evidence shown) would be just confirmative at this stage.

Change: Additional data was obtained for H1889A variant and is included in the manuscript (p11). **Supplementary Figure 27** was added to show the reaction time course of the H189A variant as compared to reaction of wild-type enzyme. The Methods were updated.

- The authors comment that the enzyme is a homodimer, but I wonder if the work presents evidence based on methods to study proteins in solution (no figure was indicated in the text – line 247). Has this conclusion been made based solely on the structural data? A better understanding of the existence of the dimer in solution with adequate methodologies and its role in the enzyme function is required.

Response: We agree with the Reviewer. At this stage, we do not have conclusive evidence on the association state of the enzyme in solution. In the method available to us (sizing by gel filtration) the enzyme elutes at an apparent mass between monomer and dimer. Our efforts to reduce protein-matrix interactions are complicated by enzyme stability. The protein dimer in the crystal structure is well defined and the interfacial area is large, so that a true dimer is predicted with almost 80% certainty. Moreover, the MD simulations show that only the protein dimer can be modeled properly and gives stable substrate binding. Of note, the requirement for the dimer is not just based on structural stability but there is also a residue (Asn52B) from the opposite subunit used to form the binding pocket for the 3-keto-phlorizin (**Supplementary Figure 31**). In conclusion, therefore, we think that the combined evidence from structure and MD simulation supports the functional enzyme homodimer, even though the gel filtration experiment is not fully conclusive.

Change: The text on p8 was rewritten to avoid definite claim on the subunit association state of AtOGE. The biochemical results are not in any way dependent on whether the enzyme is a dimer or a monomer in solution. Like explained, the structural and computational evidence strongly supports a dimer.

- In the genomic context of *Agrobacterium tumefaciens*, there was a lack of insight into how this new enzyme relates to others of the same operon. Have other enzymes of this operon already been characterized? For the complete understanding of the metabolism of natural products, this is fundamental, and even the deletion of this gene in native bacteria.

Response: We thank the Reviewer for the comment. This operon was newly discovered as part of this research. It is novel. The activities of the other enzymes are known only in the extent as we report them here. We hope to be able to report more on these enzymes and on the whole pathway in the future. For the function of the other enzymes, please recall the following. (1) The discovery of GlycDH activity with C-glycosides was the starting point for searching operons containing homologues of the enzyme. (2) The operon of *A. tumefaciens* was discovered. (3) Please see p7 of main text (together with the associated **Supplementary Figures S20 - S22**) for the characterization of the other enzymes. (4) The Discussion in the beginning wraps up these findings.

Deletion studies in the bacteria can certainly be interesting, but they are not part of this inquiry. They are clearly out of scope and we are sure that the Reviewer will agree.

REVIEWER COMMENTS

Reviewer #1 (Remarks to the Author):

This revised manuscript by Bitter et al. details a study into the cleavage of O-glucosides that occurs via an elimination mechanism following oxidation of the C3-hydroxyl in beta-glucosides by a separate enzyme.

1) The authors write: "... Reviewer that the comparison with the alpha-phosphoglucosidase GlvA was not a useful comparison. Irrespective of expert opinion regarding this point, the best way to deal with the matter is to show both enzymes. This is what we do in the revised version and the results clarify the point." I'm not sure what the above means. The authors state that the beta- and alpha- enzyme show similarly low activity with beta-substrates and suggest that this is somehow important. The authors consider their point "to be final", perhaps but I don't agree that hard conclusions can be made.

2) I'm not disputing that the enzyme accelerates cleavage of the C-O glycosidic bond, however, for publication in a top ranking journal I would have hoped for some experimental mechanistic evidence to be included in the paper rather than stating: "Our conclusion at this stage is that the catalytic factors of the AtOGE reaction warrant more detailed mechanistic investigation for clarification in a later study". The authors finish their arguments with the following: "... but as the research stands, Figure 7B and discussion are consistent and plausible", a statement with which I disagree. The proton transfer to the aryloxy leaving group shown in Figure 7B is thermodynamically uphill until the leaving group has essentially completely departed; pKa of 2,6-dihydroxyacetophenone is 8, while that for the acid imidazole is ~7.

3, 4, 5, & 6) I'm happy with the changes made here.

7) Figure 7C, the label "tautomerization" is still present in this figure. This is not a tautomerization, what is shown are two resonance structures, only electrons are moving so a double headed "resonance" arrow should be shown.

Reviewer #2 (Remarks to the Author):

The authors have addressed all of my suggestions, and I note numerous additional experiments with extensive manuscript revision in response to concerns raised by other reviewers. I remain positive about publication of this work.

Reviewer #3 (Remarks to the Author):

The authors have well addressed my concerns. I recommend the publication of the work.

Reviewer #4 (Remarks to the Author):

The work by Bitter et al. discovers a β -eliminase that performs a O-glucosides cleavage (AtOGE). They compared the structure of this new enzyme with a previous studied one; PuCGE which was implicated in the eliminative cleavage of C-glucosides. They revealed that PuCGE is able to also perform beta-elimination of both O- and C-glucosides while AtOGE is shown as specific for the O-glycosidic substrates.

As an external reviewer added after a first revision process, I was asked to only cover the newly added MD simulations, and thus this is why I would only focus on this new section of the manuscript. The MD simulations were performed to unravel substrate binding in the two mentioned enzymes. My suggestions and doubts are the following:

- My main concern:

o Applying a soft harmonic force in the alpha-carbon atoms of the enzyme is not a standard protocol of classical MD simulations and it biases the results obtained. I would recommend justifying why it was

necessary to apply this constraint and explain very well why the results are still reliable besides this problem. Is it due to a model problem of PuCGE (monomer instead of a dimer, lack of some important residues or wrongly modelled protonation states, lack of specifying some possible disulfide bond, etc.) that cannot conserve its secondary structure along the MD simulations?

- Minor revisions:

o Page 9, line 283: I would express the sentence "AtOGE loops Asp42–Val46 and Thr198–Gly211 that are disordered in the enzyme structure were used as modeled" differently. Modelled as what? I would simply go by saying "AtOGE loops Asp42–Val46 and Thr198–Gly211 that are disordered in the enzyme structure and thus, not present in the 3D structure, were modelled (see methods section)."

o Page 9, line 296: The ring puckering results of the sugars in solution should be introduced differently or at least state that the in-solution results can be found in the supplementary information. In the present form it does not seem that the in-solution results come from the same work done by the authors.

o It would be nice also to explain in a few words the reasoning behind why the conformation in the active site differs from the one found in solution in PuCGE but not in AtOGE.

o Figure 5 could be complemented with a fourth panel superimposing the three structures to have a better idea of the same kind of binding in all the systems.

o Figure 5a: the polar hydrogens of the protein residues are not depicted while in the other panels they are.

o Page 10, line 314: For a better comprehension for a non-expert reader, I would indicate in Figure 5 where the 7-hydroxyl and C6 are placed.

o Page 26, line 930: A justification is needed explaining the choosing of a dimer for the AtOGE and a monomer for the PuCGE simulations.

o Page 27, line 972: which software was used for the clustering method?

o Page 27, line 976: which software was used to calculate the ring puckering variables Q , θ and ϕ ?

o Page 27, line 978: Using different methodologies when studying the ring conformations in solution or inside the active site of the enzyme can lead to differences in the systems because of the methodologies used instead of the environment of the sugar; which is what is wanted to study in this case. I would recommend re-doing the simulation of the sugars in-solution using the same forcefield instead of the semi-empirical method. Also, I would add a sentence explaining that force fields are not the best choice to study sugar ring conformations.

Nature communications manuscript NCOMMS-23-00431-A

Response to Reviewers/Revision

05-September-2023

Please find below our detailed response to the points raised in the second review of the NCOMMS manuscript referenced above. We are grateful for the consideration of our manuscript and the constructive criticism provided. Our responses are formatted in blue color and are additionally marked with Response. We describe our position with respect to each point raised and explain the changes made upon revision. Overall, we believe to have addressed all points in an exhaustive manner.

Reviewer Comments

Response in general

Reviewers 2 and 3 were satisfied with the changes made in the first revision. For Reviewer 1, the revision appeared to have been fine for its most part. Three points remained that we address in the second revision. The Reviewer invited at the revision stage to assess the new studies by molecular dynamics simulation gave important feedback. We are grateful for the additional constructive comments. We agree in general with the remarks made and made significant effort to provide suitable response. The requested computational studies were made and the sections lacking in clarity were reworked and elaborated in better detail.

Main changes at a glance: The concerns of Reviewer 1 were addressed through response (#1) and response/manuscript update on **p17** (#2, #7). The sections concerning molecular dynamics (MD) simulations section were revised extensively, with more detail and explanations added to improve clarity as requested by Reviewer 4. In addition to that, further computational experiments were run. MD simulations were repeated without constraint (**Figure S29**) to show stability of the system. Ring puckering analysis was performed on the same level of theory as the other experiments (**Figure S30**). Last, figures representing MD simulation data were updated and the optimizations suggested by the Reviewer were applied (**Figure 5a-d**). The used software for all calculations can now be found in the Methods section (**p26-27**).

Reviewer #1 (Remarks to the Author):

This revised manuscript by Bitter et al. details a study into the cleavage of O-glucosides that occurs via an elimination mechanism following oxidation of the C3-hydroxyl in beta-glucosides by a separate enzyme.

1) The authors write: "... Reviewer that the comparison with the alpha-phosphoglucosidase GlvA was not a useful comparison. Irrespective of expert opinion regarding this point, the best way to deal with the matter is to show both enzymes. This is what we do in the revised version and the results clarify the point." I'm not sure what the above means. The authors state that the beta- and alpha- enzyme show similarly low activity with beta-substrates and suggest that this is somehow important. The authors consider their point "to be final", perhaps but I don't agree that hard conclusions can be made.

Response: We regret that our earlier response in the Response Letter was not clear as written. However, we believe that the response was clear in content, based on the additional experimental evidence presented. The comment #1 of Reviewer 1 on the original manuscript contained the implicit request to add evidence for a GH4 enzyme that shows the same stereochemical requirements on the natural substrate used (glycoside of beta configuration) as AtOGE. We agreed with the reviewer in this specific point. (We did not agree with the opinion that the comparison with the other enzyme GlvA was not a "useful comparison".) We therefore performed new experiments and added the evidence in

the revised manuscript. We concluded at this stage that the point of Reviewer 1 was dealt with in the best way possible. We therefore considered our response to point #1 to have been final. We retain this opinion/point of view. Again, we are sorry if the written response was confusing.

Reviewer 1 now writes that he/she is not sure whether “any hard conclusions can be made” based on the evidence of GH4 enzymes. The Reviewer here extends, substantially, the scope of her/his original comment which we find as problematic and wish to point this out to the Editor. The original comment was on the choice of GH4 enzyme. We insist that this point has been addressed exhaustively. Now, a number of conclusions are drawn based on the evidence obtained with the set of GH4 enzymes. The most important conclusion is that the enzymes were unable to cleave the C-glycoside nothofagin while they showed activity with the corresponding O-glycoside phloretin. Respectfully, we don’t have any use for the adjective “hard”, and the tacitly implied negative-opposite “soft”, in this discussion. Our conclusions are based on direct experimental facts without extensive interpretation of the results.

2) I'm not disputing that the enzyme accelerates cleavage of the C-O glycosidic bond, however, for publication in a top ranking journal I would have hoped for some experimental mechanistic evidence to be included in the paper rather than stating: "Our conclusion at this stage is that the catalytic factors of the AtOGE reaction warrant more detailed mechanistic investigation for clarification in a later study". The authors finish their arguments with the following: "... but as the research stands, Figure 7B and discussion are consistent and plausible", a statement with which I disagree. The proton transfer to the aryloxy leaving group shown in Figure 7B is thermodynamically uphill until the leaving group has essentially completely departed; pKa of 2,6-dihydroxyacetophenone is 8, while that for the acid imidazole is ~7.

Response: We disagree with the Reviewer in several points encapsulated in this comment #2. Most importantly, we reject the statement “*for publication in a top ranking journal I would have hoped for some experimental mechanistic evidence to be included in the paper rather than stating*”. Not only is the statement vague in content (which evidence, concretely, is the reviewer talking about here?), but it also carries the insinuation that something important was missing from our study for publication in a “top-ranking journal”. The reviewer can’t be serious in implying that our study does not contain “some experimental mechanistic evidence”. A significant body of evidence from enzyme discovery, protein structure, MD simulations, biochemical and kinetic data, and mutagenesis results are combined here to put forward what we believe are highly plausible mechanistic proposals for the catalytic reactions of CGE and OGE. We will address the point of disagreement of Reviewer 1 below. As far as we can see, the key question in relation to the above statement of Reviewer 1 is therefore this: Are the mechanistic proposals adequately supported by evidence and is anything missing from them that one could have reasonably expected in terms of level of detail and completeness, while at the same time respecting also the identity and scientific scope of the study performed? We suggest that the answer is yes to the first part and no to the second. Reviewer 1 may disagree with the position of the authors, but it may then be necessary to accept difference in opinions and leave the task of arbitration to the Editor. There is no scientific study in which it wouldn’t be possible to ask for more to be done. We can only speculate that perhaps the Reviewer is referring to studies by kinetic isotope effect (KIE) as tools of mechanistic investigation (e.g., Sannikova et al. on the GH4 mechanism; see comment #2 of Reviewer 1 on the original manuscript). While the unique power of KIEs in mechanistic inquiry is certainly beyond debate, such investigations simply did not have reasonable importance and were not of priority in the current research as designed by the authors.

The author statement from the previous Response Letter "Our conclusion at this stage is that the catalytic factors of the AtOGE reaction warrant more detailed mechanistic investigation for clarification in a later study" was meant in relation to the rate acceleration provided by AtOGE (comment #4 of

Reviewer 1 in the original review). We show in experiments that the enzyme accelerates the bond cleavage compared to the spontaneous reaction in solution. The Reviewer appears to be in agreement. We also show evidence in support of the mechanistic principles used by the enzyme in catalytic rate acceleration (assistance from Mn^{2+} cofactor, general base and general acid catalysis; the role of general acid catalysis is disputed by Reviewer 1). Our statement was to say that at this stage, we are unable to evaluate the different catalytic factors of the enzyme in terms of their relative contribution to the observed rate acceleration. We retain the position that it is reasonable to leave such investigation (effectively a new topic of specialized mechanistic interest) to a potential inquiry in the future.

Reviewer 1 disagrees with the proposed mechanism of general acid-catalyzed expulsion of the leaving group in C-O bond cleavage (the mechanism of AtOGE). The Reviewer's argument holds that the pK_a of the proposed catalytic group on the enzyme (protonated histidine) is close to the pK_a of the leaving group (phloretin) and so full proton transfer between the two groups requires that the scissile C-O bond is largely broken. The argument might be rephrased to say that general acid catalysis (proton transfer) will have a limited role, potentially none, in promoting the C-O bond cleavage in the C3'-oxidized phloretin. We have no problem with agreeing with Reviewer 1 on such general considerations of chemical reactivity, but where is the essential point in respect to our scheme/discussion? The histidine may protonate the leaving group at a late stage of the bond cleavage, but what is problematic about showing the overall reaction step (bond cleavage and proton transfer) as formally one? A scheme drawn in this way does not imply a particular degree of concertedness of the bond cleavage and the proton transfer. It is indifferent to this particular detail of the mechanism and we believe that the vast majority of chemical readers would understand it in just that way. It only indicates that these two events must happen to proceed to the product. To the authors it seems that Reviewer 1, apparently inspired by the mechanistic proposal of unassisted expulsion of leaving group in catalytic reactions of GH4 enzymes, is very much in favor of a stepwise mechanism where an anionic leaving group is released and only later becomes protonated. We agree that the stepwise reaction is a possible scenario but it is only one extreme in a continuum of equally plausible scenarios with different relative timings of bond cleavage proceeding somewhat concerted with proton transfer. First of all, just small changes in group pK_a at the enzyme active site, say, the His pK_a was lowered to 5.5 and the phloretin pK_a was elevated to 9.0 (both changes wouldn't be uncommon in enzyme active sites), would completely change the whole argument. Our point here is certainly not to select pK_a values to better suit our mechanistic proposal, but to emphasize how difficult the argument based on assumed pK_a differences can be. Second, even if the reaction was stepwise with C-O bond cleavage happening far ahead of the proton transfer, there is still the question if the enzyme really expels the negatively charged leaving group. We find it much more likely that the enzyme will avoid charge accumulation and will do so with the help of histidine. The interaction with the catalytic histidine may be reduced to a mere hydrogen bonding until the C-O bond cleavage is advanced suitably and it will eventually result in the proton to be transferred with effectively no energy barrier. The proposed role of the histidine is supported as structurally possible by the results of MD simulation. Lastly, we wish to emphasize that we do not consider more complicated proposals of enzyme mechanisms that involve a partially rate-determining protein conformational change (see the proposals for GH4 enzymes based on KIE results). Proton transfer might be associated with such conformational change. There is no evidence from our current results that would support such discussion and in line with Occam's razor we avoid it.

Changes made: We have adapted the reaction scheme of AtOGE (**Figure 7b**) so that the departure of the aryloxy leaving group and the protonation of the oxyanion are now shown as two separate consecutive steps of reaction (**p17**). We add in the legend to Figure 7b that this is an extreme case. Various scenarios of relative timing of the two steps are possible and would be consistent with the results shown. Reviewer 1 has not provided convincing argument that the stepwise reaction would be a unique possibility of mechanism.

3, 4, 5, & 6) I'm happy with the changes made here.

7) Figure 7C, the label "tautomerization" is still present in this figure. This is not a tautomerization, what is shown are two resonance structures, only electrons are moving so a double headed "resonance" arrow should be shown.

Response: We apologize for not correcting this mistake in the first revision and thank the Reviewer for pointing this out again. Our only explanation is that in the course of an extensive and lengthy revision this point was missed in the final version of the revised manuscript that was submitted. The label was changed from "tautomerization" to a double headed resonance arrow, as suggested and as should be.

Reviewer #2 (Remarks to the Author):

The authors have addressed all of my suggestions, and I note numerous additional experiments with extensive manuscript revision in response to concerns raised by other reviewers. I remain positive about publication of this work.

Reviewer #3 (Remarks to the Author):

The authors have well addressed my concerns. I recommend the publication of the work.

Reviewer #4 (Remarks to the Author):

The work by Bitter et al. discovers a β -eliminase that performs a O-glucosides cleavage (AtOGE). They compared the structure of this new enzyme with a previous studied one; PuCGE which was implicated in the eliminative cleavage of C-glucosides. They revealed that PuCGE is able to also perform beta-elimination of both O- and C-glucosides while AtOGE is shown as specific for the O-glycosidic substrates.

As an external reviewer added after a first revision process, I was asked to only cover the newly added MD simulations, and thus this is why I would only focus on this new section of the manuscript. The MD simulations were performed to unravel substrate binding in the two mentioned enzymes. My suggestions and doubts are the following:

- My main concern:

o Applying a soft harmonic force in the alpha-carbon atoms of the enzyme is not a standard protocol of classical MD simulations and it biases the results obtained. I would recommend justifying why it was necessary to apply this constraint and explain very well why the results are still reliable besides this problem. Is it due to a model problem of PuCGE (monomer instead of a dimer, lack of some important residues or wrongly modelled protonation states, lack of specifying some possible disulfide bond, etc.) that cannot conserve its secondary structure along the MD simulations?

Response: We apologize for the typo in the method part regarding the quaternary structure of PuCGE in the simulations. Both proteins were simulated as a dimer and not as a monomer (PuCGE as a heterodimer with the two subunits alpha and beta, and AtOGE as a homodimer). As an example, in Fig. 5 we showed PuCGE residues of the two subunits for the interaction with 3-keto-puerarin. According to this, the description of the MD simulation in the Methods part has been corrected into:

"The all-atom models for the binding modes of 3-keto-puerarin and 3-keto-phlorizin to PuCGE and 3-keto-phlorizin to AtOGE were obtained by a combined protocol of molecular docking and classical molecular dynamics (MD) simulations in explicit aqueous solvent. We considered the homodimer of AtOGE and the heterodimer of PuCGE for our simulations".

Regarding the use of small constrains on the alpha-carbon atoms of PuCGE in the MD simulations, we agree that imposing constraints biased the outcome of the simulation. Nevertheless, in this work we wanted to refine the Michaelis complex of PuCGE with 3-keto-puerarin and 3-keto-phlorizin, and the complex of AtOGE with 3-keto-phlorizin. No problems of stability of the system along the MD

simulations. Most of the structure of both proteins was already solved by the X-ray crystallographic model. In another words, our scientific question was not about the secondary and ternary structure of the enzyme.

In any case, and prompted by the comments of the reviewer, we have simulated all systems without any restraint into the system. We run 300 ns per simulation, 3 independent unrestrained MD simulations per system (total time of 0.9 μ s per system). Additionally, in the case of AtOGE, we improved our model by inclusion of the missing C-terminal part in the X-ray structure. All the systems were stable along the MD simulations with RMSD values for the backbone of the protein (alpha-carbons) below or equal to 1.0 Å when compared to the last snapshot of the constrained simulation and with RMSD values for the heavy atoms of the substrates very similar to the ones obtained in the constrained simulations (see new **Figure S29**). This confirms that no big conformational changes or instabilities of the systems are present.

On other hand, as stated in the Methods part, (...) “the protonation state of the side chains of His, Lys, Arg, Asp and Glu was assessed at pH 6.0 via the server H++”.

Finally, and in order to clarify the scope of why we used MD simulations in this manuscript, we have rewritten the section ‘Molecular dynamics simulations suggest substrate positioning for catalysis in AtOGE and PuCGE’ of results as well as the parts related to the MD simulations in the discussion. We hope the text is now clearer for the reader.

- Minor revisions:

o Page 9, line 283: I would express the sentence “AtOGE loops Asp42–Val46 and Thr198–Gly211 that are disordered in the enzyme structure were used as modeled” differently. Modelled as what? I would simply go by saying “AtOGE loops Asp42–Val46 and Thr198–Gly211 that are disordered in the enzyme structure and thus, not present in the 3D structure, were modelled (see methods section).”

Response: We appreciate the suggestion of the Reviewer. As we stated above, we have rewritten the MD section and this sentence has been removed from the main text. In the Methods part is written:

‘In the case of AtOGE, we modeled the two missing loops in the X-ray structure (residues Asp42-Val46 and Thr198-Gly211) using the RCD+ server.’

o Page 9, line 296: The ring puckering results of the sugars in solution should be introduced differently or at least state that the in-solution results can be found in the supplementary information. In the present form it does not seem that the in-solution results come from the same work done by the authors.

Response: We thank the Reviewer for this suggestion. In the new MD section, we have written for the puckering analysis:

‘We also monitor the puckering of the sugar moiety in 3-keto-puerarin and 3-keto-phlorizin when bound to the enzymes and in solution (Figure S30). Whereas both substrates populate in major extension a ⁴C₁ conformation in solution, the binding to the metal center promotes conformations around 4H5/5E for both substrates. Other minor conformations are also observed in 3-keto-phlorizin when bound to PuCGE.’

o It would be nice also to explain in a few words the reasoning behind why the conformation in the active site differs from the one found in solution in PuCGE but not in AtOGE.

Response: In the new simulations where the C-terminus of AtOGE has been also included, there are no significant differences for the puckering of the sugar moiety in 3-keto-phlorizin when bound to AtOGE or PuCGE.

o Figure 5 could be complemented with a fourth panel superimposing the three structures to have a better idea of the same kind of binding in all the systems.

Response: We thank the reviewer for this suggestion and have included a fourth panel in the structure main figure (**Figure 5d**). We have superimposed all three docking poses, showing the coordination and positioning of the substrate by the metal site. For simplicity, we have removed any protein side chains.

o Figure 5a: the polar hydrogens of the protein residues are not depicted while in the other panels they are.

Response: We apologize for this inconsistent depiction and have removed polar hydrogens from all protein residues in the main text figures for simplicity (**Figure 4&5**), but are therefore showing them in the SI Figures (**Figure S31, S33 and S35**).

o Page 10, line 314: For a better comprehension for a non-expert reader, I would indicate in Figure 5 where the 7-hydroxyl and C6 are placed.

Response: We thank the Reviewer for this suggestion. We have now included such labels in **Figure 5: C2' and C3' from the sugar moiety (a-d)** as well as C6 and C7 from the C-linked aglycone (c).

o Page 26, line 930: A justification is needed explaining the choosing of a dimer for the AtOGE and a monomer for the PuCGE simulations.

Response: As commented before, both enzymes have been simulated as a dimer. The typo in the methods part has been corrected.

o Page 27, line 972: which software was used for the clustering method?

Response: We used *cpptraj* in AmberTools21. This has been included into the Methods section:

'A representative structure of each of the substrate:enzyme (ES) complexes was selected from the larger cluster of conformers out of three computed with the hierarchical agglomerative (bottom-up) algorithm available in cpptraj.'

o Page 27, line 976: which software was used to calculate the ring puckering variables Q, theta and phi?

Response: The puckering variables were calculated with the program *geo.py* from the software SHARC (sharc-md.org). This has been included into the Methods part:

'In order to evaluate the effect of the active site environment on the puckering of the glycan part of the ligands, we measured the Cremer-Pople puckering descriptors Q, theta and phi on both trajectories of the ligands in solution and bound to the metal center at the active site using the program geo.py included into SHARC (<https://sharc-md.org>).'

o Page 27, line 978: Using different methodologies when studying the ring conformations in solution or inside the active site of the enzyme can lead to differences in the systems because of the methodologies used instead of the environment of the sugar; which is what is wanted to study in this

case. I would recommend re-doing the simulation of the sugars in-solution using the same forcefield instead of the semi-empirical method. Also, I would add a sentence explaining that force fields are not the best choice to study sugar ring conformations.

Response: We agree with the Reviewer that the use of several levels of theory not for all cases can be misleading. We have simplified the message and we have simulated, as suggested, both substrates 3-keto-phlorizin and 3-keto-puerarin in solution at the same level of theory (force field) with explicit solvation (OPC water molecules). The outcome of these results can be seen in the new **Figure S30**.

REVIEWERS' COMMENTS

Reviewer #2 (Remarks to the Author):

Reviewer 1 makes two points about the scientific arguments in the manuscript which I will provide my opinion on below, as well as pointing out a third mistake that then is corrected by the author without further comment and so will not be addressed here.

1) This point appears to be a continued discussion of a comment with the same number made in the first round of reviews. In that first round of review, Reviewer 1 made a comment that a "comparison with a GH4 enzyme using a non-natural diastereomeric substrate with the enzyme AtOGE using its natural substrate is, in my opinion, not a useful comparison. A more useful comparison would have been to a beta-phosphoglucosidase". The authors conceded this point, and so expressed and tested an additional GH4 enzyme BglT with a natural diastereomeric substrate that matches their newly discovered AtOGE, while still retaining in their manuscript the data measured with the original comparison enzyme GlvA.

I am of the opinion that all the data should be shown, including the original comparison with GlvA, and that the reader can draw their own conclusions on how useful this comparison is. I do not feel that this comparison is over-interpreted, and I consider the original critique is then addressed.

In the second round of reviews, Reviewer 1 adds "The authors state that the beta- and alpha- enzyme show similarly low activity with beta-substrates and suggest that this is somehow important". The authors could consider making the relevance of their conclusions from this comparison more explicit to address this.

2) Reviewer 1 states a hope for experimental mechanistic evidence. It is unclear to me what specific evidence this should be, and a 'hope' is then in my interpretation not a requirement. It certainly would be nice to see even more experiments dissecting the mechanism, but this can be a long process and for other enzymes a whole body of literature together is needed to give a 'hard' answer to this question. There are many small details that can be investigated to make strong claims about an enzyme's mechanism, such as the exact nature of the transition state and extent of bond formation and breaking within this (or multiple transition states in this case, as the enzyme catalyses multiple steps). As the author points out, some of this could be answered by kinetic isotope effects, but this would require extensive synthesis of isotope-labeled substrates and in the opinion of the author would be another study entirely. I agree with this assessment.

Reviewer 1 has not specified an experiment that is necessary, and so in my opinion their hopes for more mechanistic evidence should not be a hurdle to publication.

Reviewer 1 here further argues from the relative pKa values of the catalytic residue and the leaving group that the protonation step will be thermodynamically uphill, and so is likely not relevant to the mechanism as written at the time of initial submission. However, pKa values for side-chains inside an enzyme active site can deviate dramatically from those of a simple amino acid in solution, and so to me this argument does not hold. Taking the two catalytic carboxylates in inverting glycosidases as an example, the catalytic acid often has a pKa 5-6 log units away from the pKa of the free amino acid side chain and is protonated at neutral pH. It is not at all implausible to me that the protonated His in the active site also has a pKa that is perturbed from that of a normal imidazole, and so is able to transfer a proton to the leaving group. However, as there is not any direct evidence for either option the authors have adjusted the figure to present this as a two-step process of leaving group departure followed by protonation, with a caveat in the caption that the exact sequence of steps is not yet known.

This adjustment of the figure and caption in my mind addresses this question.

Overall I then consider both critiques to be adequately addressed, with the only potential further adjustment needed being a small clarification to make explicit the relevance of the comparisons with GH4 enzymes.

Reviewer #4 (Remarks to the Author):

The authors did a good job addressing my concerns, so I think their work should be published.

Nature communications manuscript NCOMMS-23-00431-A

Response to Reviewers/Revision

05-Oct-23

Please find below our detailed response to the points raised in the final review of the NCOMMS manuscript referenced above. We are grateful for the consideration of our manuscript and the constructive criticism provided. Our responses are formatted in blue color and are additionally marked with Response. We describe our position with respect to each point raised and explain the changes made upon revision. Overall, we believe to have addressed all points in an exhaustive manner.

Reviewer Comments

Response in general

We are grateful for positive remarks by Reviewer #2 and #4.

Reviewer #2 (Remarks to the Author):

Reviewer 1 makes two points about the scientific arguments in the manuscript which I will provide my opinion on below, as well as pointing out a third mistake that then is corrected by the author without further comment and so will not be addressed here.

This point appears to be a continued discussion of a comment with the same number made in the first round of reviews. In that first round of review, Reviewer 1 made a comment that a “comparison with a GH4 enzyme using a non-natural diastereomeric substrate with the enzyme AtOGE using its natural substrate is, in my opinion, not a useful comparison. A more useful comparison would have been to a beta-phosphoglucosidase”. The authors conceded this point, and so expressed and tested an additional GH4 enzyme BglT with a natural diastereomeric substrate that matches their newly discovered AtOGE, while still retaining in their manuscript the data measured with the original comparison enzyme GlvA.

I am of the opinion that all the data should be shown, including the original comparison with GlvA, and that the reader can draw their own conclusions on how useful this comparison is. I do not feel that this comparison is over-interpreted, and I consider the original critique is then addressed.

Response: We thank the reviewer for the positive assessment of the revisions made. It is reassuring to hear this opinion after the lengthy debate with Reviewer 1.

In the second round of reviews, Reviewer 1 adds “The authors state that the beta- and alpha-enzyme show similarly low activity with beta-substrates and suggest that this is somehow important”. The authors could consider making the relevance of their conclusions from this comparison more explicit to address this.

Response: We are of the opinion that the sub-section “Family GH4 glycoside hydrolases do not cleave phloretin C-glucoside” (page 9 of revised manuscript) is clear as it stands. We use the heading to indicate the main point (not known before). To address the issue of anomeric selectivity that created so extensive debate in the review, we write (with emphasis added in yellow background): “We chose two well-characterized GH4 enzymes (GlvA from *Bacillus subtilis*⁵⁵; BglT from *Thermotoga maritima*³²). Both enzymes require a 6-phospho hexopyranoside as substrate. They differ in preference for the substrate anomeric configuration (α for GlvA, β for BglT), yet can utilize α - and β -

configured glucosides with good leaving groups.” Now, phloretin and daidzein are relatively good leaving groups. It follows just from this fact alone that testing the two enzymes has made sense. We add another sentence to make explicit the point.

2) Reviewer 1 states a hope for experimental mechanistic evidence. It is unclear to me what specific evidence this should be, and a ‘hope’ is then in my interpretation not a requirement. It certainly would be nice to see even more experiments dissecting the mechanism, but this can be a long process and for other enzymes a whole body of literature together is needed to give a ‘hard’ answer to this question. There are many small details that can be investigated to make strong claims about an enzyme’s mechanism, such as the exact nature of the transition state and extent of bond formation and breaking within this (or multiple transition states in this case, as the enzyme catalyses multiple steps). As the author points out, some of this could be answered by kinetic isotope effects, but this would require extensive synthesis of isotope-labeled substrates and in the opinion of the author would be another study entirely. I agree with this assessment.

Response: It is reassuring to have this opinion of Reviewer #2.

Reviewer 1 has not specified an experiment that is necessary, and so in my opinion their hopes for more mechanistic evidence should not be a hurdle to publication.

Response: We are grateful for this support.

Reviewer 1 here further argues from the relative pKa values of the catalytic residue and the leaving group that the protonation step will be thermodynamically uphill, and so is likely not relevant to the mechanism as written at the time of initial submission. However, pKa values for side-chains inside an enzyme active site can deviate dramatically from those of a simple amino acid in solution, and so to me this argument does not hold. Taking the two catalytic carboxylates in inverting glycosidases as an example, the catalytic acid often has a pKa 5-6 log units away from the pKa of the free amino acid side chain and is protonated at neutral pH. It is not at all implausible to me that the protonated His in the active site also has a pKa that is perturbed from that of a normal imidazole, and so is able to transfer a proton to the leaving group. However, as there is not any direct evidence for either option the authors have adjusted the figure to present this as a two-step process of leaving group departure followed by protonation, with a caveat in the caption that the exact sequence of steps is not yet known.

This adjustment of the figure and caption in my mind addresses this question.

Response: We are grateful for the agreement with our line of arguments and the revisions made.

Overall I then consider both critiques to be adequately addressed, with the only potential further adjustment needed being a small clarification to make explicit the relevance of the comparisons with GH4 enzymes.

Response: We thank the reviewer for the positive response. We revised the text as follows (the full paragraph is shown to indicate the context. Changes are highlighted in yellow background):

“Both enzymes require a 6-phospho hexopyranoside as substrate. They differ in preference for the substrate anomeric configuration (α for GlvA, β for BglT), yet can utilize α - and β -configured glucosides with good leaving groups.³³ The 6-phospho derivatives of phlorizin and nothofagin were synthesized (**Figures S58-S59**) and assessed for reactivity with GlvA and BglT, based on measurement

of aglycone release. Note that phloretin is a relatively good leaving group, rendering both GlvA and BgIT as interesting candidates to be evaluated.”

Reviewer #4 (Remarks to the Author):

The authors did a good job addressing my concerns, so I think their work should be published.

Response: We thank the reviewer for the positive assessment of the revision.